# Learning protein structure with a differentiable simulator

**John Ingraham[1][*], Adam Riesselman[1], Chris Sander[1,2,3], Debora Marks[1,3]**
[1]Harvard Medical School     [2]Dana-Farber Cancer Institute
[3]Broad Institute of Harvard and MIT

## Abstract

The Boltzmann distribution is a natural model for many systems, from brains to materials and biomolecules, but is often of limited utility for fitting data because Monte Carlo algorithms are unable to simulate it in available time. This gap between the expressive capabilities and sampling practicalities of energy-based models is exemplified by the protein folding problem, since energy landscapes underlie contemporary knowledge of protein biophysics but computer simulations are challenged to fold all but the smallest proteins from first principles. In this work we aim to bridge the gap between the expressive capacity of energy functions and the practical capabilities of their simulators by using an unrolled Monte Carlo simulation as a model for data. We compose a neural energy function with a novel and efficient simulator based on Langevin dynamics to build an end-to-end-differentiable model of atomic protein structure given amino acid sequence information. We introduce techniques for stabilizing backpropagation under long roll-outs and demonstrate the model's capacity to make multimodal predictions and to, in some cases, generalize to unobserved protein fold types when trained on a large corpus of protein structures.

## 1 Introduction

Many natural systems, such as cells in a tissue or atoms in a protein, organize into complex structures from simple underlying interactions. Explaining and predicting how macroscopic structures such as these arise from simple interactions is a major goal of science and, increasingly, machine learning.

The Boltzmann distribution is a foundational model for relating local interactions to system behavior, but can be difficult to fit to data. Given an energy function $U_{\boldsymbol{\theta}}[\boldsymbol{x}]$, the probability of a system configuration $\boldsymbol{x}$ scales exponentially with energy as

$$p_{\boldsymbol{\theta}}(\boldsymbol{x}) = \frac{1}{Z}\exp\left(-U_{\boldsymbol{\theta}}[\boldsymbol{x}]\right), \qquad (1)$$

where the (typically intractable) constant $Z$ normalizes the distribution. Importantly, simple energy functions $U_{\boldsymbol{\theta}}[\boldsymbol{x}]$ consisting of weak, local interactions can collectively encode complex system behaviors, such as the structures of materials and molecules or, when endowed with latent variables, the statistics of images, sound, and text (Ackley et al., 1985; Salakhutdinov & Larochelle, 2010). Unfortunately, learning model parameters $\hat{\boldsymbol{\theta}}$ and generating samples $\boldsymbol{x} \sim p_{\boldsymbol{\theta}}(\boldsymbol{x})$ of the Boltzmann distribution is difficult in practice, as these procedures depend on expensive Monte Carlo simulations that may struggle to mix effectively. These difficulties have driven a shift towards generative models that are easier to learn and sample from, such as directed latent variable models and autoregressive models (Goodfellow et al., 2016).

The *protein folding problem* provides a prime example of both the power of energy-based models at describing complex relationships in data as well as the challenge of generating samples from them. Decades of research in biochemistry and biophysics support an energy landscape theory of

---

[*]Present address: Massachusetts Institute of Technology

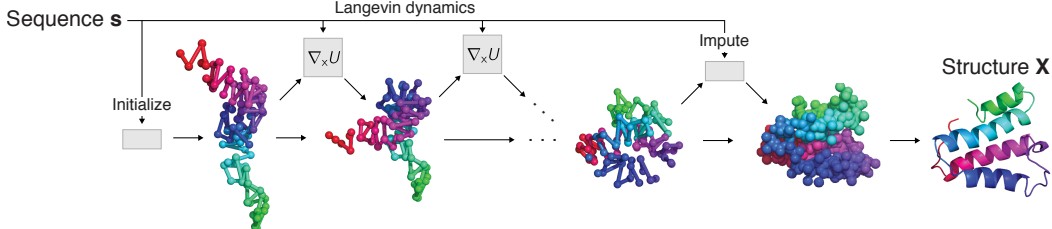

Figure 1: **An unrolled simulator as a model for protein structure.** NEMO combines a neural energy function for coarse protein structure, a stochastic simulator based on Langevin dynamics with learned (amortized) initialization, and an atomic imputation network to build atomic coordinate output from sequence information. It is trained end-to-end by backpropagating through the *unrolled* folding simulation.

protein folding (Dill et al., 2017), in which the folds that natural protein sequences adopt are those that minimize free energy. Without the availability of external information such as coevolutionary information (Marks et al., 2012) or homologous structures (Martí-Renom et al., 2000) to constrain the energy function, however, contemporary simulations are challenged to generate globally favorable low-energy structures in available time.

How can we get the representational benefits of energy-based models with the sampling efficiency of directed models? Here we explore a potential solution of directly training an *unrolled simulator* of an energy function as a model for data. By directly training the sampling process, we eschew the question 'when has the simulator converged' and instead demand that it produce a useful answer in a fixed amount of time. Leveraging this idea, we construct an end-to-end differentiable model of protein structure that is trained by *backpropagtion through folding* (Figure 1). NEMO (Neural energy modeling and optimization) can learn at scale to generate 3D protein structures consisting of hundreds of points directly from sequence information. Our main contributions are:

- **Neural energy simulator model** for protein structure that composes a deep energy function, unrolled Langevin dynamics, and an atomic imputation network for an end-to-end differentiable model of protein structure given sequence information
- **Efficient sampling algorithm** that is based on a *transform integrator* for efficient sampling in transformed coordinate systems
- **Stabilization techniques for long roll-outs** of simulators that can exhibit chaotic dynamics and, in turn, exploding gradients during backpropagation
- **Systematic analysis of combinatorial generalization** with a new dataset of protein sequence and structure

## 1.1 RELATED WORK

**Protein modeling**  Our model builds on a long history of coarse-grained modeling of protein structure (Kolinski et al., 1998; Kmiecik et al., 2016). Recently, multiple groups have demonstrated how to learn full force fields using likelihood-based approaches (Jumper et al., 2018; Krupa et al., 2017), similar to our maximum likelihood loss (but without *backpropagtion through folding* for fast sampling). While this work was in progress, two groups reported neural models of protein structure (AlQuraishi, 2018; Anand & Huang, 2018), where the former focused on modeling structure in terms of backbone angles and the latter in terms of residue-residue distances. We show how an energy function provides a natural framework to integrate both kinds of constraints, which in turn is important for achieving sample-efficient structural generalization.

**Learning to infer or sample**  Structured prediction includes a long history of casting predictions in terms of energy minimization (LeCun et al., 2006). Recently, others have built hybrid neural networks that use differentiable optimization as a building block in neural architectures (Wang et al.,

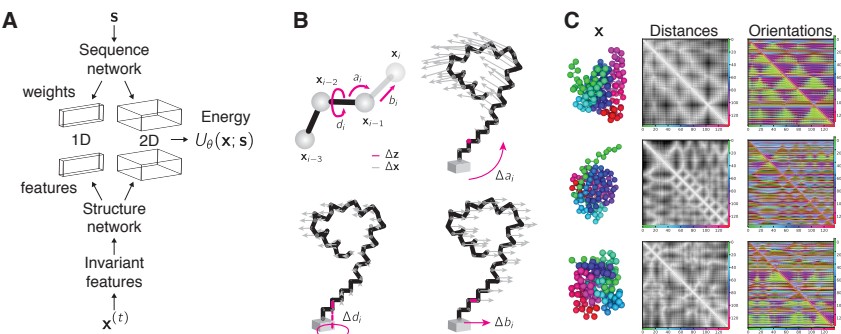

Figure 2: **A neural energy function models coarse grained structure and is sampled by internal coordinate dynamics.** (A) The energy function is formulated as a Markov Random Field with structure-based features and sequence-based weights computed by neural networks (Figure 6). (B) To rapidly sample low-energy configurations, the Langevin dynamics simulator leverages both (i) an internal coordinate parameterization, which is more effective for global rearrangements, and (ii) a Cartesian parameterization, which is more effective for localized structural refinement. (C) The base features of the structure network are rotationally and translationally invariant internal coordinates (not shown), pairwise distances, and pairwise orientations.

2016; Amos & Kolter, 2017; Belanger & McCallum, 2016). Structured Prediction Energy Networks (SPENs) with unrolled optimization (Belanger et al., 2017) are a highly similar approach to ours, differing in terms of the use of optimization rather than sampling. Additional methodologically related work includes approaches to learn energy functions and samplers simultaneously (Kim & Bengio, 2016; Wang & Liu, 2017; Dai et al., 2017; Song et al., 2017; Chen et al., 2018a), to learn efficient MCMC operators (Song et al., 2017; Levy et al., 2018), to build expressive approximating distributions with unrolled Monte Carlo simulations (Salimans et al., 2015; Titsias, 2017), and to learn the parameters of simulators with implicitly defined likelihoods[1] (Mohamed & Lakshminarayanan, 2016; Tran et al., 2017).

## 2   MODEL

**Overview**   NEMO is an end-to-end differentiable model of protein structure $X$ conditioned on sequence information $s$ consisting of three components (Figure 1): (i) a *neural energy function* $U_\theta[x; s]$ for coarse grained structure $x$ given sequence, (ii) an *unrolled simulator* that generates approximate samples from $U$ via internal coordinate Langevin dynamics (§ 2.3), and (iii) an *imputation network* that generates an atomic model $X$ from the final coarse-grained sample $x^{(T)}$ (§ 2.4). All components are trained simultaneously via backpropagation through the unrolled process.

### 2.1   REPRESENTATION

**Proteins**   Proteins are linear polymers (sequences) of amino acids that fold into defined 3D structures. The 20 natural amino acids have a common monomer structure [−(N−H)−(C−R)−(C=O)−] with variable side-chain R groups that can differ in properties such as hydrophobicity, charge, and ability to form hydrogen bonds. When placed in solvent (such as water or a lipid membrane), interactions between the side-chains, backbone, and solvent drive proteins into particular 3D configurations ('folds'), which are the basis for understanding protein properties such as biochemical activity, ligand binding, and interactions with drugs.

**Coordinate representations**   We predict protein structure $X$ in terms of 5 positions per amino acid: the four heavy atoms of the backbone (N, $C_\alpha$, and carbonyl C=O) and the center of mass of

---

[1]We leverage a traditional estimator of the likelihood gradient (Boltzmann learning) rather than ratio-based estimators

the side chain R group. While it is well-established that the locations of $C_\alpha$ carbons are sufficient to reconstruct a full atomic structure (Kmiecik et al., 2016), we include these additional positions for evaluating backbone hydrogen bonding (secondary structure) and coarse side-chain placement. Internally, the differentiable simulator generates an initial coarse-grained structure (1-position-per-amino-acid) with the loss function targeted to the midpoint of the $C_\alpha$ carbon and the side chain center of mass.

**Sequence conditioning** We consider two modes for conditioning our model on sequence information: (1) **1-seq**, in which $s$ is an $L \times 20$ matrix containing a one-hot encoding of the amino acid sequence, and (2) **Profile**, in which $s$ is an $L \times 40$ matrix encoding both the amino acid sequence and a profile of evolutionarily related sequences (§ B.7).

**Internal coordinates** In contrast to Cartesian coordinates $x$, which parameterize structure in terms of *absolute* positions of points $x_i \in \mathbb{R}^3$, internal coordinates $z$ parameterize structure in terms of *relative* distances and angles between points. We adopt a standard convention for internal coordinates of chains (Parsons et al., 2005) where each point $x_i$ is placed in a spherical coordinate system defined by the three preceding points $x_{i-1}, x_{i-2}, x_{i-3}$ in terms of a radius (bond length[2]) $b_i \in (0, \infty)$, a polar angle (bond angle) $a_i \in [0, \pi)$, and an azimuthal angle (dihedral angle) $d_i \in [0, 2\pi)$ (Figure 2B). We define $z_i = \{\tilde{b}_i, \tilde{a}_i, d_i\}$, where $\tilde{b}_i, \tilde{a}_i$ are unconstrained parameterizations of $b_i$ and $a_i$ (§ A.1). The transformation $x = \mathcal{F}(z)$ from internal coordinates to Cartesian is then defined by the recurrence

$$x_i = x_{i-1} + b_i \left[ \hat{u}_{i-1} \ \ \hat{n}_{i-1} \times \hat{u}_{i-1} \ \ \hat{n}_{i-1} \right] \begin{bmatrix} \cos(\pi - a_i) \\ \sin(\pi - a_i)\cos(d_i) \\ \sin(\pi - a_i)\sin(d_i) \end{bmatrix},$$

where $\hat{u}_i = \frac{x_i - x_{i-1}}{\|x_i - x_{i-1}\|}$ is a unit vector from $x_{i-1}$ to $x_i$ and $\hat{n}_i = \frac{\hat{u}_{i-1} \times \hat{u}_i}{\|\hat{u}_{i-1} \times \hat{u}_i\|}$ is a unit vector normal to each bond plane. The inverse transformation $z = \mathcal{F}^{-1}(x)$ is simpler to compute, as it only involves local (and fully parallelizable) calculations of distances and angles (§ A.1).

## 2.2 NEURAL ENERGY FUNCTION

**Deep Markov Random Field** We model the distribution of a structure $x$ conditioned on a sequence $s$ with the Boltzmann distribution, $p_\theta(x|s) = \frac{1}{Z}\exp\left(-U_\theta[x; s]\right)$, where $U_\theta[x; s]$ is a sequence-conditioned energy function parameterized by a neural network. Our approach is compatible with any differentiable energy function $U[x; s]$, though we focus on a decomposition

$$U_\theta[x; s] = \sum_i l_i(s; \theta) f_i(x; \theta), \tag{2}$$

which is a Markov Random Field with coefficients $\{l_i(s; \theta)\}_{i=1}^{M}$ computed by a *sequence network* and structural features $\{f_i(x; \theta)\}_{i=1}^{M}$ computed by a *structure network* (Figure 2A). This decomposition facilitates (i) increased interpretability, as the (learned) structural features are independent of sequence, and (ii) increased computational efficiency, as the sequence-based coefficients can be computed once and reused throughout a simulation.

**Sequence network** The sequence network takes as input one-dimensional sequence information $s$ and outputs: (1) *Energetic coefficients*, a set of 1- and 2-dimensional sequence features $\{l_i(s; \theta)\}_{i=1}^{M}$, (2) *Simulator initial state* $z^{(0)}$, (3) *Simulator hyperparameters* preconditioning matrix $C$, and (4) *Predicted secondary structure* (Figure 6). It is parameterized by a combination of 1D, 2D, and graph convolutions (Gilmer et al., 2017) (§ A).

**Structure network** The structure network takes as input a coarse-grained structure $x$ and outputs a set of 1D and 2D structural features $\{f_i(x; \theta)\}_{i=1}^{M}$ (Figure 6). We design the energy function to be invariant to rigid body motions (rotations and translations in SE(3)) by leveraging a set of invariant base features (Figure 2C) which are:

---

[2]Since our representation $x$ is coarse-grained at one point per position, these are virtual bonds.

1. **Internal coordinates** $z$ All internal coordinates except 6 are invariant to rotation and translation[3] and we mask these in the energy loss.

2. **Distances** $D_{ij} = \|x_i - x_j\|$ between all pairs of points. We further process these by 4 radial basis functions with (learned) Gaussian kernels.

3. **Orientation vectors** $\hat{v}_{ij}$, which are unit vectors encoding the relative position of point $x_j$ in a local coordinate system of $x_i$ with base vectors $\frac{\hat{u}_i - \hat{u}_{i+1}}{\|\hat{u}_i - \hat{u}_{i+1}\|}$, $\hat{n}_{i+1}$, and the cross product thereof.

## 2.3 Efficient simulator

**Langevin dynamics** The Langevin dynamics is a stochastic differential equation that aymptotically samples from the Boltzmann distribution (Equation 1). It is typically simulated by a first-order discretization as

$$x^{(t+\epsilon)} \leftarrow x^{(t)} - \frac{\epsilon}{2}\nabla_x U^{(t)} + \sqrt{\epsilon}\mathbf{p}, \qquad \mathbf{p} \sim \mathcal{N}(0, I). \tag{3}$$

**Internal coordinate dynamics** The efficiency with which Langevin dynamics explores conformational space is highly dependent on the geometry (and thus parameterization) of the energy landscape $U(x)$. While Cartesian dynamics are efficient at local structural rearrangement, internal coordinate dynamics much more efficiently sample global, coherent changes to the topology of the fold (Figure 2B) . We interleave the Cartesian Langevin dynamics with preconditioned Internal Coordinate dynamics,

$$z^{(t+\epsilon)} \leftarrow z^{(t)} - \frac{\epsilon C}{2}\nabla_z U^{(t)} + \sqrt{\epsilon C}\mathbf{p}, \qquad \mathbf{p} \sim \mathcal{N}(0, I), \tag{4}$$

where $C$ is a preconditioning matrix that sets the relative scaling of changes to each degree of freedom. For all simulations we unroll $T = 250$ time steps, each of which is comprised of one Cartesian step followed by one internal coordinate step (Equation 9,§ A.3).

**Transform integrator** Simulating internal coordinate dynamics is often computationally intensive as it requires rebuilding Cartesian geometry $x$ from internal coordinates $z$ with $\mathcal{F}(z)$ (Parsons et al., 2005) which is an intrinsically sequential process. Here we bypass the need for recomputing coordinate transformations at every step by instead computing on-the-fly *transformation integration* (Figure 3). The idea is to directly apply coordinate updates in one coordinate system to another by numerically integrating the Jacobian. This can be favorable when the Jacobian has a simple structure, such as in our case where it requires only distributed cross products.

## 2.4 Atomic imputation

**Local reference frame reconstruction** The *imputation network* builds an atomic model $X$ from the final coarse coordinates $x^{(T)}$. Each atomic coordinate $X_{i,j}$ of atom type $j$ at position $i$ is placed in a local reference frame as

$$X_{i,j} = x_i + e_{i,j}(z;\theta)\left[\hat{u}_i \ \hat{n}_{i+1} \ \hat{n}_{i+1} \times \hat{u}_i\right]r_{i,j}(z;\theta),$$

where $e_{i,j}(z;\theta)$ and $r_{i,j}(z;\theta)$ are computed by a 1D convolutional neural network (Figure 6).

## 3 Training

We train and evaluate the model on a set of ~67,000 protein structures (domains) that are hierarchically and temporally split. The model is trained by gradient descent using a composite loss that combines terms from likelihood-based and empirical-risk minimization-based training.

---

[3]Specifically, the six-degrees of freedom parameterizing the rigid body placement of the structure are the 'virtual' $\{b_1, a_1, a_2, d_1, d_2, d_3\}$.

| **Algorithm 1:** Direct integrator | **Algorithm 2:** Transform integrator |
|---|---|
| **Input** : State $\boldsymbol{z}^{(0)}$, energy $U(\boldsymbol{x})$, step $\epsilon$, time $T$, scale $\mathbf{C}$
**Output** : Trajectory $\boldsymbol{x}^{(0)}, \dots, \boldsymbol{x}^{(T)}$
Initialize $\boldsymbol{x}^{(0)} \leftarrow \mathcal{F}(\boldsymbol{z}^{(0)})$;
**while** $t < T$ **do**
  Compute forces $\boldsymbol{f_z} = -\frac{\partial \boldsymbol{x}}{\partial \boldsymbol{z}}^T \nabla_{\boldsymbol{x}} U$;
  Sample $\Delta \boldsymbol{z} \sim \mathcal{N}\left(\frac{1}{2}\epsilon\mathbf{C}\boldsymbol{f_z}, \epsilon\mathbf{C}\right)$;
  ■ $\boldsymbol{z}^{(t+\epsilon)} \leftarrow \boldsymbol{z}^{(t)} + \Delta\boldsymbol{z}$;
  ■ $\boldsymbol{x}^{(t+\epsilon)} \leftarrow \mathcal{F}(\boldsymbol{z}^{(t+\epsilon)})$;
  $t \leftarrow t + \epsilon$;
**end** | **Input** : State $\boldsymbol{z}^{(0)}$, energy $U(\boldsymbol{x})$, step $\epsilon$, time $T$, scale $\mathbf{C}$
**Output** : Trajectory $\boldsymbol{x}^{(0)}, \dots, \boldsymbol{x}^{(T)}$
Initialize $\boldsymbol{x}^{(0)} \leftarrow \mathcal{F}(\boldsymbol{z}^{(0)})$;
**while** $t < T$ **do**
  Compute forces $\boldsymbol{f_z} = -\frac{\partial \boldsymbol{x}}{\partial \boldsymbol{z}}^T \nabla_{\boldsymbol{x}} U$;
  Sample $\Delta \boldsymbol{z} \sim \mathcal{N}\left(\frac{1}{2}\epsilon\mathbf{C}\boldsymbol{f_z}, \epsilon\mathbf{C}\right)$;
  ■ $\tilde{\mathbf{x}} \leftarrow \boldsymbol{x}^{(t)} + \frac{\partial \boldsymbol{x}}{\partial \boldsymbol{z}}^{(t)} \Delta\boldsymbol{z}^{(t)}$;
  ■ $\boldsymbol{x}^{(t+\epsilon)} \leftarrow \boldsymbol{x}^{(t)} + \frac{1}{2}\left(\frac{\partial \boldsymbol{x}}{\partial \boldsymbol{z}}^{(t)} + \frac{\partial \tilde{\mathbf{x}}}{\partial \boldsymbol{z}}\right)\Delta\boldsymbol{z}^{(t)}$;
  $t \leftarrow t + \epsilon$;
**end** |

Figure 3: A transform integrator simulates Langevin dynamics in a more favorable coordinate system (e.g. internal coordinates $\mathbf{z}$) directly in terms of the untransformed state variables (e.g. Cartesian $\mathbf{x}$). This exchanges the cost of an inner-loop transformation step (e.g. geometry construction $\mathcal{F}(\boldsymbol{z})$) for an extra Jacobian evaluation, which is fully parallelizable on modern hardware (e.g. GPUs).

## 3.1 DATA

**Structural stratification**  There are several scales of generalization in protein structure prediction, which range from predicting the structure of a sequence that differs from the training set at a few positions to predicting a 3D fold topology that is absent from training set. To test these various levels of generalization systematically across many different protein families, we built a dataset on top of the CATH hierarchical classification of protein folds (Orengo et al., 1997). **CATH** hierarchically organizes proteins from the Protein Data Bank (Berman et al., 2000) into domains (individual folds) that are classified at the levels of **C**lass, **A**rchitecture, **T**opology, and **H**omologous superfamily (from general to specific). We collected protein domains from CATH releases 4.1 and 4.2 up to length 200 and hierarchically and temporally split this set (§ B.1) into training (~35k folds), validation (~21k folds), and test sets (~10k folds).

**Test subsets**  The final test set is subdivided into four subsets: **C**, **A**, **T**, and **H**, based on the level of maximal similarity between a given test domain and domains in the training set. For example, domains in the **C** or **A** sets may share class and potentially architecture classifications with train but will not share topology (i.e. fold).

## 3.2 LOSS

**Likelihood**  The gradient of the data-averaged log likelihood of the Boltzmann distribution is

$$\frac{\partial}{\partial \theta_i} \mathbb{E}_{\boldsymbol{x} \sim \text{Data}} \left[\log p(\boldsymbol{x} | \boldsymbol{s}, \theta)\right] = \mathbb{E}_{\boldsymbol{x} \sim p(\boldsymbol{x}|\theta)} \left[\frac{\partial}{\partial \theta_i} U_\theta(\boldsymbol{x}; \boldsymbol{s})\right] - \mathbb{E}_{\boldsymbol{x} \sim \text{Data}} \left[\frac{\partial}{\partial \theta_i} U_\theta(\boldsymbol{x}; \boldsymbol{s})\right], \quad (5)$$

which, when ascended, will minimize the average energy of samples from the data relative to samples from the model. In an automatic differentiation setting, we implement a Monte Carlo estimator for (the negative of) this gradient by adding the *energy gap*,

$$\mathcal{L}_{\text{ML}} = U_\theta(\bot(\boldsymbol{x}^{(\text{D})}); \boldsymbol{s}) - U_\theta(\bot(\boldsymbol{x}^{(\text{M})}); \boldsymbol{s}), \quad (6)$$

to the loss, where $\bot$ is an identity operator that sets the gradient to zero[4].

**Empirical Risk**  In addition to the likelihood loss, which backpropagates through the energy function but not the whole simulation, we developed an empirical risk loss composing several measures of protein model quality. It takes the form

$$\mathcal{L}_{\text{ER}} = \mathcal{L}_{\text{Distances}} + \mathcal{L}_{\text{Angles}} + \mathcal{L}_{\text{H-bonds}} + \mathcal{L}_{\text{TM-score}} + \mathcal{L}_{\text{Init}} + \mathcal{L}_{\text{Trajectory}} \quad (7)$$

---

[4]In TensorFlow this operation is `stop_gradient`.

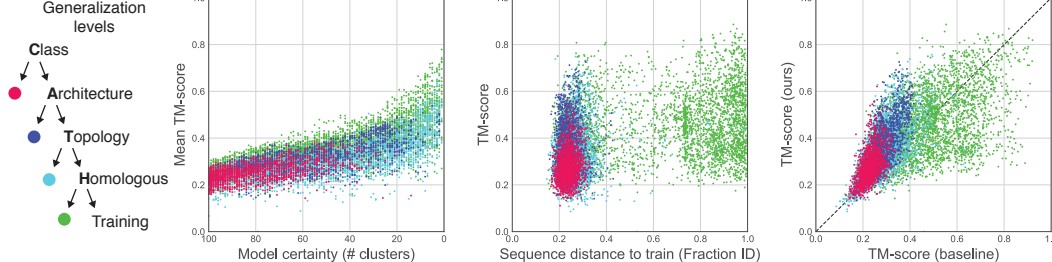

Figure 4: **Model generalizes and outperforms end-to-end baseline for unseen fold topologies.**
Colors indicate varying difficulty levels of protein domains in the test set, with the **C** (cyan) and **A**
(magenta) subsets containing corresponding to test-set domains with topologies (folds) and super-
families that were not represented in the training set. (Left) As the model exhibits higher confidence
(reduced structural diversity), it becomes more accurate. (Center) The model occasionally achieves
TM scores greater than 0.5 even for difficult **C** and **A** level generalization tasks. (Right) NEMO
outperforms a strong RNN baseline for difficult generalization problems. All results for NEMO and
RNN baselines are conditioned on profiles.

Table 1: Test set performance across different levels of generalization

| Model | # params | Total | C | A | T | H |
|---|---|---|---|---|---|---|
| NEMO (ours, profile) | 21.3m | **0.366** | **0.274** | **0.361** | **0.331** | 0.431 |
| NEMO (ours, sequence-only) | 19.1m | 0.248 | 0.198 | 0.245 | 0.254 | 0.263 |
| RNN baseline model (profile) | | | | | | |
| 2x100 | 5.9m | 0.293 | 0.213 | 0.230 | 0.247 | 0.388 |
| 2x300 (avg. of 3) | 8.8m | 0.335 | 0.229 | 0.282 | 0.278 | 0.446 |
| 2x500 | 13.7m | 0.347 | 0.222 | 0.272 | 0.286 | **0.477** |
| 2x700 | 21.4m | 0.309 | 0.223 | 0.259 | 0.261 | 0.403 |
| Number of structures | | 10381 | 1537 | 1705 | 3198 | 3941 |

schematized in Figure 6. Our combined loss sums all of the terms $\mathcal{L} = \mathcal{L}_{\mathrm{ER}} + \mathcal{L}_{\mathrm{ML}}$ without weighting.

### 3.3 STABILIZING BACKPROPAGATION THROUGH TIME

We found that the long roll-outs of our simulator were prone to chaotic dynamics and exploding
gradients, as seen in other work (Maclaurin et al., 2015; Parmas et al., 2018). Unfortunately, when
chaotic dynamics do occur, it is typical for *all* gradients to explode (across learning steps) and
standard techniques such as gradient clipping (Pascanu et al., 2013) are unable to rescue learning
(§ B.5). To stabilize training, we developed two complimentary techniques that regularize against
chaotic simulator dynamics while still facilitating learning when they arise. They are

- **Lyapunov regularization** We regularize the simulator time-step function (rather than the
  energy function) to be approximately 1-Lipschitz. (If exactly satisfied, this eliminates the
  possibility of chaotic dynamics.)

- **Damped backpropagation through time** We exponentially decay gradient accumulation
  on the backwards pass of automatic differentiation by multiplying each backwards iteration
  by a damping factor $\gamma$. We adaptively tune $\gamma$ to cancel the scale of the exploding gradients.
  This can be thought of as a continuous relaxation of and a quantitatively tunable alternative
  to truncated backpropagation through time.

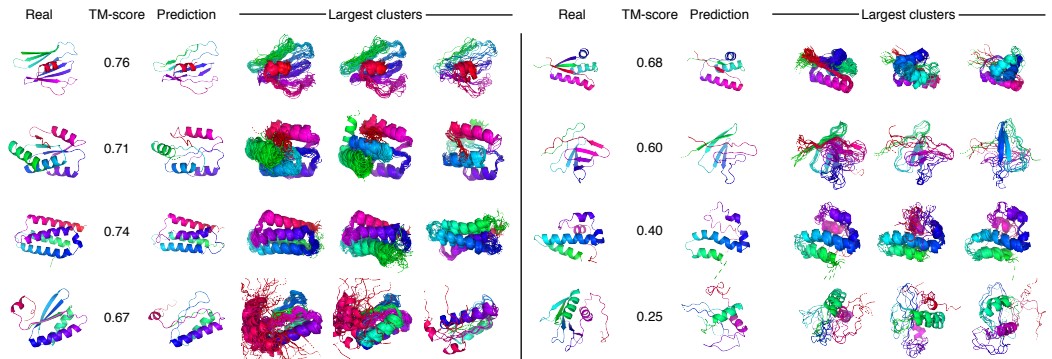

Figure 5: **Examples of fold generalization at topology and architecture level**. These predicted structures show a range of prediction accuracy for structural generalization (**C** and **A**) tasks, with the TM-score comparing the top ranked 3D-Jury pick against the target. The largest clusters are the three most-populated clusters derived from 100 models per domain with a within-cluster cutoff of TM > 0.5. CATH IDs: 2oy8A03; 5c3uA02; 2y6xA00; 3cimB00; 4ykaC00; 2f09A00; 3i5qA02; 2ayxA01.

## 4 RESULTS

### 4.1 GENERALIZATION ACROSS CATH

For each of the 10,381 protein structures in our test set, we sampled 100 models from NEMO, clustered them by structural similarity, and selected a representative structure by a standard consensus algorithm (Ginalski et al., 2003). For evaluation of performance we focus on the TM-Score (Zhang & Skolnick, 2005), a measure of structural similarity between 0 and 1 for which TM > 0.5 is typically considered an approximate reconstruction of a fold.

**Calibrated uncertainty**   We find that, when the model is confident (i.e. the number of distinct structural clusters is low ∼1-3), it is also accurate with some predictions having average TM > 0.5 (Figure 4, left). Unsurprisingly, the confidence of the model tends to go with the difficulty of generalization, with the most confident predictions from the **H** test set and the least confident from **C**.

**Structural generalization**   However, even when sequence identity is low and generalization difficulty is high (Figure 4, center), the model is still able to make some accurate predictions of 3D structures. Figure 5 illustrates some of these successful predictions at **C** and **A** levels, specifically 4ykaC00, 5c3uA02 and beta sheet formation in 2oy8A03. We observe that the predictive distribution is multimodal with non-trivial differences between the clusters representing alternate packing of the chain. In some of the models there is uneven distribution of uncertainty along the chain, which sometimes corresponded to loosely packed regions of the protein.

**Comparison to an end-to-end baseline**   We constructed a baseline model that is a non-iterative replica of NEMO which replaces the coarse-grained simulator module (and energy function) with a two-layer bidirectional LSTM that directly predicts coarse internal coordinates $z^{(0)}$ (followed by transformation to Cartesian coordinates with $\mathcal{F}$). We trained this baseline across a range of hyperparameter values and found that for difficult **C**, **A**, and **T** tasks, NEMO generalized more effectively than the RNNs (Table 1). For the best performing 2x300 architecture, we trained two additional replicates and report the averaged perfomance in Figure 4 (right).

Additionally, we report the results of a sequence-only NEMO model in Table 1. Paralleling secondary structure prediction (Rost & Sander, 1993; McGuffin et al., 2000), we find that the availability of evolutionary information has significant impact on prediction quality.

## 4.2 ADVANTAGES AND DISADVANTAGES

This work presents a novel approach for protein structure prediction that combines the inductive bias of simulators with the speed of directed models. A major advantage of the approach is that model sampling (inference) times can be considerably faster than conventional approaches to protein structure prediction (Table 4). There are two major disadvantages. First, the computational cost of training and sampling is higher than that of angle-predicting RNNs (Figure 10) such as our baseline or AlQuraishi (2018). Consequently, those methods have been scaled to larger datasets than ours (in protein length and diversity) which are more relevant to protein structure prediction tasks. Second, the instability of backpropagating through long simulations is unavoidable and only partially remedied by our approaches of Lipschitz regularization and gradient damping. These approaches can also lead to slower learning and less expressive energy functions. Methods for efficient (i.e. subquadratic) $N$-body simulations and for more principled stabilization of deep networks may be relevant to addressing both of these challenges in the future.

## 5 CONCLUSION

We described a model for protein structure given sequence information that combines a coarse-grained neural energy function and an unrolled simulation into an end-to-end differentiable model. To realize this idea at the scale of real proteins, we introduced an efficient simulator for Langevin dynamics in transformed coordinate systems and stabilization techniques for backpropagating through long simulator roll-outs. We find that that model is able to predict the structures of protein molecules with hundreds of atoms while capturing structural uncertainty, and that the model can structurally generalize to distant fold classifications more effectively than a strong baseline.

## ACKNOWLEDGEMENTS

We thank members of the Marks lab for useful discussions and feedback. Parts of this work were performed on the Orchestra compute cluster at Harvard Medical School.

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

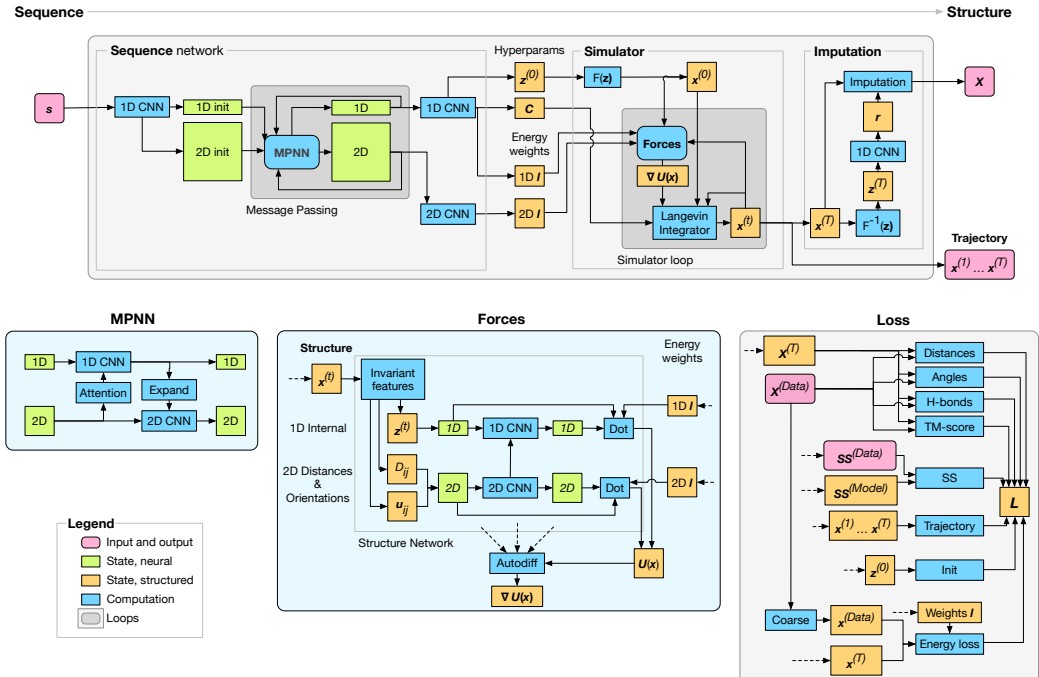

Figure 6: **Model schematic**. The model generates an atomic structure $\boldsymbol{X}$ (top right) from sequence information $\boldsymbol{s}$ (top left) via 3 steps: First, a sequence network takes in the sequence information $\boldsymbol{s}$, processes it with a combination of 1D, 2D, and graph convolutions (MPNN, bottom left), and outputs energy function weights $l$ as well as simulator hyperparameters (top center). Second, the simulator iteratively modifies the structure via Langevin dynamics based on the gradient of the energy landscape (Forces, bottom center). Third, the imputation network constructs predicted atomic coordinates $\boldsymbol{X}$ from the final simulator time step $\boldsymbol{x}^{(T)}$. During training, the true atomic coordinates $\boldsymbol{X}^{\text{(Data)}}$, predicted atomic coordinates $\boldsymbol{X}$, simulator trajectory $\boldsymbol{x}^{(1)}, \ldots, \boldsymbol{x}^{(T)}$, and secondary structure predictions $SS^{\text{(Model)}}$ feed into a composite loss function (Loss, bottom right), which is then optimized via backpropagation.

# APPENDICES

# A MODEL

## A.1 COORDINATE SYSTEMS

**Inverse transformation** The inverse transformation $\boldsymbol{z} = \mathcal{F}^{-1}(\boldsymbol{x})$ involves fully local computations of bong lengths and angles.

$$b_i = ||\boldsymbol{x}_i - \boldsymbol{x}_{i-1}||, \quad a_i = \arccos\left(-\hat{\boldsymbol{u}}_i \cdot \hat{\boldsymbol{u}}_{i-1}\right), \quad d_i = \text{sign}\left(\hat{\boldsymbol{u}}_{i-2} \cdot \hat{\boldsymbol{n}}_i\right) \arccos\left(\hat{\boldsymbol{n}}_{i-1} \cdot \hat{\boldsymbol{n}}_i\right).$$

**Jacobian** The Jacobian $\frac{\partial \boldsymbol{x}}{\partial \boldsymbol{z}}$ defines the infinitesimal response of the Cartesian coordinates $\mathbf{x}$ to perturbations of the internal coordinates $\mathbf{z}$. It will be important for both converting Cartesian forces into angular torques and bond forces as well as the development of our transform integrator. It is

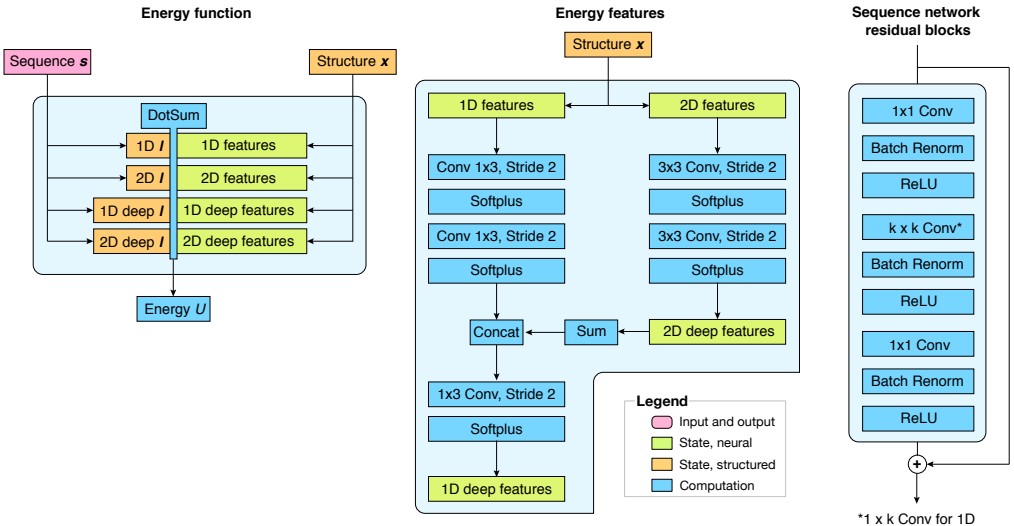

Figure 7: **Component architectures**. (Left) The energy function is the inner product of sequence-based weights and structure-based features. A combination of low- and high-level features capture multi-scale constraints on structure. (Center) The structure network is a lightweight convolutional network operating on both 1D (backbone) and 2D (interaction) features. (Right) Convolutional neural network modules used for sequence processing are composed of residual blocks that interleave spatial convolutions with 1x1 convolutions.

defined element-wise as

$$\frac{\partial \boldsymbol{x}_j}{\partial b_i} = \begin{cases} \hat{\boldsymbol{u}}_i & i \le j \\ 0 & i > j \end{cases},$$

$$\frac{\partial \boldsymbol{x}_j}{\partial a_i} = \begin{cases} \hat{\boldsymbol{n}}_i \times (\boldsymbol{x}_j - \boldsymbol{x}_{i-1}) & i \le j \\ 0 & i > j \end{cases},$$

$$\frac{\partial \boldsymbol{x}_j}{\partial d_i} = \begin{cases} \hat{\boldsymbol{u}}_{i-1} \times (\boldsymbol{x}_j - \boldsymbol{x}_{i-1}) & i \le j \\ 0 & i > j \end{cases}.$$

The Jacobian has a simple form that can be understood by imagining the protein backbone as a robot arm that is planted at $\boldsymbol{x}_0$ (Figure 2B). Increasing or decreasing the **bond length** $b_i$ extends or retracts all downstream coordinates along the bonds axis, moving a **bond angle** $a_i$ drives circular motion of all downstream coordinates around the bond normal vector $\hat{\boldsymbol{n}}_i$ centered at $\boldsymbol{x}_{i-1}$, and moving a **dihedral angle** $d_i$ drives circular motion of downstream coordinate $\boldsymbol{x}_j$ around bond vector $\hat{\boldsymbol{u}}_{i-1}$ centered at $\boldsymbol{x}_{i-1}$.

**Unconstrained representations**  Bond lengths and angles are subject to the constraints $b_i > 0$ and $0 < a_i < \pi$. We enforce these constraints by representing these degrees of freedom in terms of fully unconstrained variables $\tilde{b}_i$ and $\tilde{a}_i$ via the transformations $b_i = \log\left(1 + e^{\tilde{b}_i}\right)$ and $a_i = \frac{\pi}{1+e^{-\tilde{a}_i}}$. All references to the internal coordinates $\boldsymbol{z}$ and Jacobians $\frac{\partial \boldsymbol{x}}{\partial \boldsymbol{z}}$ will refer to the use of fully unconstrained representations (Table 2).

## A.2  ENERGY FUNCTION

Figure 6 provides an overall schematic of the model, including the components of the energy function.

**CNN primitives**  All convolutional neural network primitives in the model schematic (Figure 6) follow a common structure consisting of stacks of residual blocks. Each residual block includes

Table 2: Coordinate systems and representations for protein structure.

| Variable | Notation | Shape |
|---|---|---|
| Sequence | $s$ | [L, 20] |
| Cartesian coordinates (coarse) | $x$ | [3L,1] |
| Internal coordinates | $z$ | [3L,1] |
| Cartesian coordinates (atomic) | $\mathbf{X}$ | [3L,A] |
| Cartesian coordinates for position $i$ | $x_i$ | [3,1] |
| Internal coordinate for position $i$ | $z_i = \begin{bmatrix} b_i\ \tilde{a}_i\ \tilde{d}_i \end{bmatrix}^T$ | [3,1] |
| Unit vector from $x_{i-1}$ to $x_i$ | $\hat{u}_i$ | [3,1] |
| Unit vector normal to bond plane at $x_{i-1}$ | $\hat{n}_i$ | [3,1] |
| Bond length $\|x_i - x_{i-1}\|$ | $b_i$ | [1] |
| Bond angle $\angle(\hat{u}_i, -\hat{u}_{i-1})$ | $a_i$ | [1] |
| Dihedral angle $\angle(\hat{n}_i, \hat{n}_{i-1})$ | $d_i$ | [1] |
| Unconstrained bond length | $\tilde{b}_i$ | [1] |
| Unconstrained bond angle | $\tilde{a}_i$ | [1] |
| Jacobian matrix | $\frac{\partial x}{\partial z}$ | [3L,3L] |

consists of a layer of channel mixing (1x1 convolution), a variable-sized convolution layer, and a second layer of channel mixing. We use dropout with $p = 0.9$ and Batch Renormalization (Ioffe, 2017) on all convolutional layers. Batch Renormalization rather than Normalization was necessary rather owing to the large variation in sizes of the structures of the proteins and resulting large variation in mini-batch statistics.

## A.3  INTERNAL COORDINATE DYNAMICS WITH A TRANSFORM INTEGRATOR

**Why sampling vs. optimization**   Deterministic methods for optimizing the energy $U(x; s)$ such as gradient descent or quasi-Newton methods can effectively seek local minima of the energy surface, but are challenged to optimize globally and completely ignore the contribution of the widths of energy minima (entropy) to their probability. We prefer sampling to optimization for three reasons: (i) noise in sampling algorithms can facilitate faster global conformational exploration by overcoming local minima and saddle points, (ii) sampling generates populations of states that respect the width (entropy) of wells in $U$ and can be used for uncertainty quantification, and (iii) sampling allows training with an approximate Maximum Likelihood objective (Equation 5).

**Langevin Dynamics**   The Langevin dynamics are a stochastic dynamics that sample from the canonical ensemble. They are defined as a continuous-time stochastic differential equation, and are simulated in discrete time with the first order discretization

$$x^{(t+\epsilon)} \leftarrow x^{(t)} - \frac{\epsilon}{2}\nabla_x U^{(t)} + \sqrt{\epsilon}\mathbf{p}, \qquad \mathbf{p} \sim \mathcal{N}(0, I). \tag{8}$$

Each time step of $\epsilon$ involves a descent step down the energy gradient plus a perturbation of Gaussian noise. Importantly, as time tends toward to infinity, the time-distribution of the Langevin dynamics converges to the canonical ensemble. Our goal is to design a dynamics that converge to an approximate sample in a very short period of time.

Table 3: **Model architecture**. Input number of channels $q = 20$ for sequence-only and $q = 40$ for profiles.

| Location | Type | Channels | # Blocks | Width | Dilation | Stride |
|---|---|---|---|---|---|---|
| Pre-MPNN | 1D | 128 | 12 | 3 | [1, 2, 4, 8] × 3 | 1 |
| MPNN | 1D | 128 | 4 | 3 | [1, 2, 4, 8] | 1 |
| MPNN | 2D | 50 | 1 | 7 | 1 | 1 |
| Post-MPNN | 1D | q+256 | 12 | 3 | [1, 2, 4, 8] × 3 | 1 |
| Post-MPNN* | 2D | 100 | 1 | 9 | 1 | 1 |
| Imputation | 1D | q+256 | 12 | 3 | [1, 2, 4, 8] × 3 | 1 |

**Coordinate systems and preconditioning**    The efficiency with which Langevin dynamics explore conformational space is highly dependent on the geometry of the energy landscape $U(\boldsymbol{x})$, which in turn depends on how the system is parameterized. Molecular energy functions in Cartesian coordinates tend to exhibit strong correlations between variables that result from the requirement that underlying molecular geometries satisfy highly stereotyped bond lengths and angles. As a result, simulations of naive Cartesian Langevin dynamics require a small time step to satisfy these constraints and tend to be dominated by high-frequency, localized vibrations of the chain. The large, global motions that are essential to protein folding can require thousands to millions of times steps to manifest.

A well-known solution to the complex dependencies of Cartesian coordinates is to carry out optimization and simulation in internal coordinates, which directly parameterize molecular geometries in terms of the bond lengths and angles (Parsons et al., 2005). Internal coordinate parameterizations possess the advantages that (i) bond length and angle constraints are easy to satisfy and (ii) small changes to a single angle can drive large, coherent rearrangements of the chain (Figure 2B). For example, simply replacing $\mathbf{x}$'s with $\mathbf{z}$'s in Equation 8 yields the dynamics

$$\boldsymbol{z}^{(t+\epsilon)} \leftarrow \boldsymbol{z}^{(t)} - \frac{\epsilon}{2}\nabla_{\boldsymbol{z}}U^{(t)} + \sqrt{\epsilon}\mathbf{p}, \qquad \mathbf{p} \sim \mathcal{N}(0, I).$$

The advantages and disadvantages of the two coordinate systems are complementary: Cartesian dynamics efficiently sample local structural rearrangements and inefficiently sample global chain motions, while internal coordinate dynamics efficiently sample global, *correlated* motions of the chain but are challenged to make precise local rearrangements.

The time dynamics of these alternative parameterizations need not be kinetically realistic to converge to the correct distribution over conformational space. Different coordinate systems warp the local geometry of the energy landscape and will in turn rescale and redirect which global vibrational and local vibrations dominate the dynamics. This relative rescaling can be further optimized by applying a global linear transformation to the energy landscape with a preconditioning 'inverse mass' matrix $\boldsymbol{C}$, giving the update

$$\boldsymbol{z}^{(t+\epsilon)} \leftarrow \boldsymbol{z}^{(t)} - \frac{\epsilon\boldsymbol{C}}{2}\nabla_{\boldsymbol{z}}U^{(t)} + \sqrt{\epsilon\boldsymbol{C}}\mathbf{p}, \qquad \mathbf{p} \sim \mathcal{N}(0, I). \tag{9}$$

**Transform integrator**    The need to rebuild Cartesian geometry $\boldsymbol{x}$ from internal coordinates $\boldsymbol{z}$ with $\mathcal{F}(\boldsymbol{z})$ at every time step is one of the major costs of conformational sampling codes based on Internal coordinates (Parsons et al., 2005) because it is intrinsically sequential. Here we show how it is possible to bypass the need for geometry reconstruction at every step by instead computing on-the-fly *geometry modification*.

Imagine following a change to the internal coordinates $\Delta\boldsymbol{z}^{(t)}$ along a straight path from $\boldsymbol{z}^{(t)}$ to $\boldsymbol{z}^{(t+\epsilon)}$ and tracking the corresponding nonlinear path of the Cartesian coordinates from $\boldsymbol{x}^{(t)}$ to $\boldsymbol{x}^{(t+\epsilon)}$. If this path is indexed by $u \in (t, t + \epsilon)$, then the dynamics of $\mathbf{x}$ with respect to $u$ are given by $\frac{\partial \boldsymbol{x}}{\partial u} = \frac{\partial \boldsymbol{x}}{\partial \boldsymbol{z}}\frac{\partial \boldsymbol{z}}{\partial u} = \frac{\partial \boldsymbol{x}}{\partial \boldsymbol{z}}\frac{1}{\epsilon}\Delta\boldsymbol{z}$. Integrating the dynamics of $\mathbf{x}$ gives

$$\boldsymbol{x}^{(t+\epsilon)} = \mathcal{F}\left(\boldsymbol{z}^{(t)} + \Delta\boldsymbol{z}^{(t)}\right)$$

$$= \boldsymbol{x}^{(t)} + \int_{t}^{t+\epsilon} \frac{1}{\epsilon}\frac{\partial \boldsymbol{x}}{\partial \boldsymbol{z}}^{(u)}\Delta\boldsymbol{z}^{(t)}du.$$

This illustrates that it is possible to convert coordinate changes in one coordinate system (e.g. Internal Coordinates) to coordinate changes in another (e.g. Cartesian) by integrating an autonomous system of ODEs with dynamics governed by the Jacobian. Since $\epsilon$ is small, we integrate this system with a single step of Heun's method (improved Euler), where we first substitute an Euler approximation to predict $\boldsymbol{x}^{(t+\epsilon)}$ as

$$\tilde{\mathbf{x}}^{(t+\epsilon)} \approx \boldsymbol{x}^{(t)} + \frac{\partial \boldsymbol{x}}{\partial \boldsymbol{z}}^{(t)}\Delta\boldsymbol{z}^{(t)},$$

and then substitute the Jacobian evaluated at the predicted state $\tilde{\mathbf{x}}^{(t+\epsilon)}$ to form trapezoidal approximation

$$\boldsymbol{x}^{(t+\epsilon)} \approx \boldsymbol{x}^{(t)} + \frac{1}{2}\left(\frac{\partial \boldsymbol{x}}{\partial \boldsymbol{z}}^{(t)} + \frac{\partial \tilde{\mathbf{x}}}{\partial \boldsymbol{z}}^{(t+\epsilon)}\right)\Delta\boldsymbol{z}^{(t)}.$$

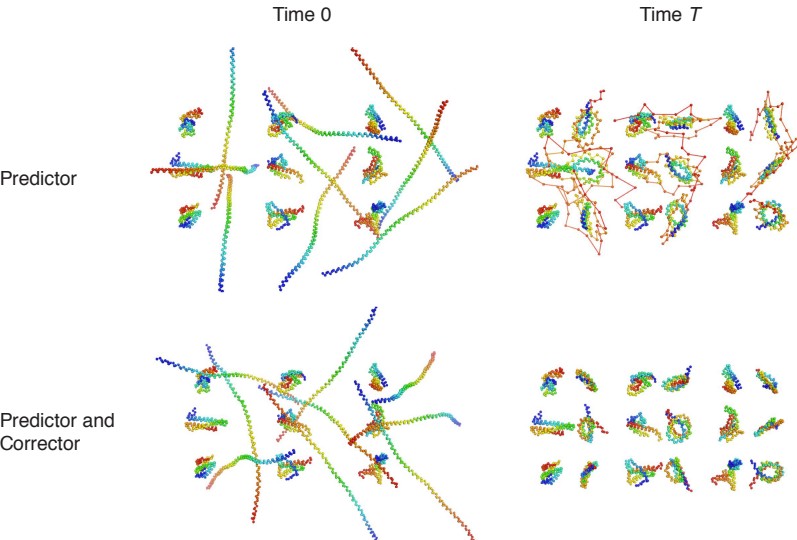

Figure 8: **Accounting for second order errors is essential for internal coordinate dynamics.** (Top) Discarding the corrector step rapidly accumulates errors due to the curvilinear motions of internal coordinate dynamics. (Bottom) Heun integration with a corrector step accounts for curvature in curvilinear motion.

The comparison of this algorithm with naive integration is given in Figure 8. The corrector step is important for eliminating the large second-order errors that arise in curvilinear motions caused by angle changes (Figure 2B and Figure 8). In principle higher-order numerical integration methods or more time steps could increase accuracy at the cost of more evaluations of the Jacobian, but we found that second-order effects seemed to be the most relevant on our timescales.

**Mixed integrator**    Cartesian dynamics favor local structural rearrangements, such as the transitioning from a helical to an extended conformation, while internal coordinate dynamics favor global motions such as the change of the overall fold topology. Since both kinds of structural rearrangements are important to the folding process, we form a hybrid integrator (Algorithm 3) by taking one step with each integrator per force evaluation.

**Translational and rotational detrending**    Both Cartesian and Internal coordinates are overparameterized with $3L$ degrees of freedom, since only $3L - 6$ degrees of freedom are necessary to encode a centered and un-oriented structure[5]. As a consequence, a significant fraction of the per time-step changes $\Delta \boldsymbol{x}$ can be explained by rigid translational and rotational motions of the entire structure. We isolate and remove these components of motion by treating the system $\{\boldsymbol{x}_1, \dots, \boldsymbol{x}_L\}$ as a set of particles with unit mass, and computing effective structural translational and rotational velocities by summing point-wise momenta.

The translational component of motion is simply the average displacement across positions $\Delta \boldsymbol{x}_i^{\text{Trans}} = \langle \Delta \boldsymbol{x}_i \rangle$. For rotational motion around the center of mass, it is convenient to define the non-translational motion as $\Delta \bar{\boldsymbol{x}}_i = \Delta \boldsymbol{x}_i - \Delta \boldsymbol{x}_i^{\text{Trans}}$ and the centered Cartesian coordinates as $\bar{\boldsymbol{x}}_i = \boldsymbol{x}_i - \langle \boldsymbol{x}_i \rangle$. The point-wise angular momentum is then $\boldsymbol{l}_i = \bar{\boldsymbol{x}}_i \times \Delta \bar{\boldsymbol{x}}_i$ and we define a total angular velocity of the structure $\boldsymbol{\omega}$ by summing these and dividing by the moment of inertia as $\boldsymbol{\omega} = \left( \sum_i \boldsymbol{l}_i \right) / \left( \sum_i \|\bar{\boldsymbol{x}}_i\|_2^2 \right)$. We convert the angular velocity $\boldsymbol{\omega}$ into Cartesian displacements with an unrolled Heun integration as $\Delta \boldsymbol{x}_i^{\text{Rot}} = \frac{1}{2} \boldsymbol{\omega} \times (\bar{\boldsymbol{x}}_i + \boldsymbol{\omega} \times \bar{\boldsymbol{x}}_i)$, which leaves the isolated structural motions as $\Delta \boldsymbol{x}_i^{\text{Struct}} = \Delta \boldsymbol{x}_i - \Delta \boldsymbol{x}_i^{\text{Trans}} - \Delta \boldsymbol{x}_i^{\text{Rot}}$.

---

[5]In Internal coordinates the rotation and translation of the structure are encoded in $b_1, a_1, a_2, d_1, d_{,2}, d_3$ while in Cartesian coordinates they are distributed across all coordinates.

---

**Algorithm 3:** Mixed Integrator

---

**Input** : Initial state $\boldsymbol{z}^{(0)}$, energy $U(\boldsymbol{x})$, time steps $\epsilon_{\boldsymbol{x}}, \epsilon_{\boldsymbol{z}}$, total time $T$, preconditioners $\mathbf{C}_{\boldsymbol{x}}, \mathbf{C}_{\boldsymbol{z}}$,
**Output** : Trajectory $\boldsymbol{x}^{(0)}, \ldots, \boldsymbol{x}^{(T)}$
Initialize $\boldsymbol{x}^{(0)} \leftarrow \mathcal{F}(\boldsymbol{z}^{(0)})$;
**while** $t < T$ **do**
$\quad \boldsymbol{f}_{\boldsymbol{x}} \leftarrow \nabla_{\boldsymbol{x}} U$;
$\quad \Delta \boldsymbol{x}^{(Cart)} \leftarrow \texttt{CartesianStep}(\boldsymbol{x}^{(t)}, \boldsymbol{f}_{\boldsymbol{x}}, \epsilon_{\boldsymbol{x}}, \mathbf{C}_{\boldsymbol{x}})$;
$\quad \Delta \boldsymbol{x}^{(Int)} \leftarrow \texttt{ClippedInternalStep}(\boldsymbol{x} + \Delta \boldsymbol{x}^{(Cart)}, \boldsymbol{f}_{\boldsymbol{x}}, \epsilon_{\boldsymbol{z}}, \mathbf{C}_{\boldsymbol{z}})$;
$\quad \boldsymbol{x} \leftarrow \boldsymbol{x} + \texttt{Detrend}(\Delta \boldsymbol{x}^{(Cart)} + \Delta \boldsymbol{x}^{(Int)})$;
$\quad t \leftarrow t + \epsilon$;
**end**

---

**Speed clipping**  We found it helpful to stabilize the model by enforcing a speed limit on overall structural motions for the internal coordinate steps. This prevents small changes to the energy function during learning from causing extreme dynamics that in turn produce a non-informative learning signal. To accomplish this, we translationally and rotationally detrend the update of the predictor step $\Delta \boldsymbol{x}$ and compute a hypothetical time step $\hat{\epsilon}_{\boldsymbol{z}}$ that would limit the fastest motion to 2 Angstroms per iteration. We then compute modified predictor and corrector steps subject to this new, potentially slower, time step. While this breaks the asymptotics of Langevin dynamics, (i) it is unlikely on our timescales that we achieve stationarity and (ii) it can be avoided by regularizing the dynamics away from situations where clipping is necessary. In the future, considering non-Gaussian perturbations with kinetic energies similar to Relativistic Monte Carlo (Lu et al., 2017) might accomplish a similar goal in a more principled manner. The final integrator combining these ideas is presented in Figure 3.

## B   APPENDIX B: TRAINING

### B.1   DATA

For a training and validation set, we downloaded all protein domains of length $L \leq 200$ from **C**lasses $\alpha$, $\beta$, and $\alpha/\beta$ in CATH release 4.1 (2015), and then hierarchically purged a randomly selected set of **A**, **T**, and **H** categories. This created three validation sets of increasing levels of difficulty: **H**, which contains domains with superfamilies that are excluded from train (but fold topologies may be present), **T**, which contains fold topologies that were excluded from train (fold generalization), and **A** which contains secondary structure architectures that were excluded from train.

For a test set, we downloaded all folds that were new to CATH release 4.2 (2017), which (due to a propensity of structural biology to make new structures of previously solved folds), provided 10,381 test domains. We further stratified this test set into **C**, **A**, **T**, and **H** categories based on their nearest CATH classification in the training set.

We also analyzed test set stratifications based on nearest neighbors in both training and validation in figure Figure 12. We note that the validation set was not explicitly used to tune hyperparameters due to the large cost of training ( 2 months on 2 M40 GPUs), but we did keep track of validation statistics during training.

### B.2   SGD

We optimized all models for 200,000 iterations with Adam (Kingma & Ba, 2014).

### B.3   LOSS

We optimize the model using a composite loss containing several terms, which are detailed as follows.

**Distance loss**  We score distances in the model with a contact-focused distance loss

$$\sum_{i<j} w_{ij} \left| D_{ij}^{(\text{Model})} - D_{ij}^{(\text{Data})} \right|,$$

where the contact-focusing weights are

$$w_{ij} = \frac{\sigma\left(\alpha(D_0 - \min(D_{ij}^{(\text{Model})}, D_{ij}^{(\text{Data})}))\right)}{\sum_{k<l} \sigma\left(\alpha(D_0 - \min(D_{kl}^{(\text{Model})}, D_{kl}^{(\text{Data})}))\right)}$$

and $\sigma(u) = \frac{1}{1+\exp(-u)}$ is the sigmoid function.

**Angle loss**  We use the loss

$$\mathcal{L}_{\text{angles}} = \sum_i ||\mathcal{H}(z_i^{(T)}) - \mathcal{H}(z_i^{(\text{Data})})||,$$

where $\mathcal{H}(z) = [\cos(a_i) \ \sin(a_i)\cos(d_i) \ \sin(a_i)\sin(d_i)]^T$ are unit length feature vectors that map the angles $\{a_i, d_i\}$ to the unit sphere.

Other angular losses, such as the negative log probability of a Von-Mises Fisher distribution, are based on the inner product of the feature vectors $\mathcal{H}(z_a) \cdot \mathcal{H}(z_b)$ rather than the Euclidean distance $||\mathcal{H}(z_a) - \mathcal{H}(z_b)||$ between them. It is worth noting that these two quantities are directly related by $||\mathcal{H}(z_a) - \mathcal{H}(z_b)|| = \sqrt{2(1 - \mathcal{H}(z_a) \cdot \mathcal{H}(z_b))}$. Taking $z_a$ as fixed and $z_b$ as the argument, the Euclidean loss has a cusp at $z_a$ whereas the Von-Mises Fisher loss is smooth around $z_a$. This is analogous to the difference between $L^1$ and $L^2$ losses, where the cusped $L^1$ loss favors median behavior while the smooth $L^2$ loss favors average behavior.

**Trajectory loss**  In a further analogy to reinforcement learning, damped backpropation through time necessitates an intermediate loss function that can criticize transient states of the simulator. We compute this by featurizing the per time step coordinates as the product $D_{ij}\hat{v}_{ij}$ (Figure 2C) and doing the same contact-weighted averaging as the distance loss.

**Template Modelling (TM) Score**  The TM-score (Zhang & Skolnick, 2005),

$$\sum_i \frac{1}{1 + \left(\frac{D_i}{D_0}\right)^2},$$

is a measure of superposition quality between two protein structures on $[0, 1]$ that was presented as an approximately length-independent alternative to RMSD. The TM-score is the best attainable value of the preceding quantity for all possible superpositions of two structures, where $D_i = ||x^{(\text{Model})} - x^{(\text{Data})}||$. This requires iterative optimization, which we implemented with a sign gradient descent with 100 iterations to optimally superimpose the model and target structure. We backpropagate through this unrolled optimization process as well as that of the simulator.

**Hydrogen bond loss**  We determine intra-backbone hydrogen bonds using the electrostatic model of DSSP (Kabsch & Sander, 1983). First, we place virtual hydrogens at 1 Angstroms along the negative angle bisector of the $C_{i-1} - N_i - C\alpha_i$ bond angle. Second, we compute a putative energy $U_{ij}^{\text{h-bond}}$ (in kcal/mol) for each potential hydrogen bond from an amide donor at $i$ to a carbonyl acceptor at $j$ as

$$U_{ij}^{\text{h-bond}}(\mathbf{X}) = \left(\frac{q_N q_O}{D_{NO}} + \frac{q_H q_C}{D_{HC}} + \frac{q_H q_O}{D_{HO}} + \frac{q_N q_C}{D_{NC}}\right) 332$$

$$= 0.084 \left(\frac{1}{D_{NO}} + \frac{1}{D_{HC}} - \frac{1}{D_{HO}} - \frac{1}{D_{NC}}\right) 332$$

where $D_{ab} = ||\mathbf{X}_{i,a} - \mathbf{X}_{j,b}||$ is the Euclidean distance between atom $a$ of residue $i$ and atom $b$ of residue $j$. We then make hard assignments of hydrogen bonds for the data with

$$y_{ij}^{\text{data}} = \mathbf{1}\left(U_{ij}^{\text{h-bond}}(\mathbf{X}^{(\text{data})}) < -0.5\right).$$

We 'predict' the probabilities of hydrogen bonds of the data given the model via logisitic regression of soft model assignments as

$$y_{ij}^{\text{model}} = \sigma \left( a\sigma \left( b \left( -U_{ij}^{\text{h-bond}}(\mathbf{X}^{(model)}) + 0.5 \right) \right) + c \right),$$

where $a, b, c$ are learned parameters with the softplus parameterizations enforcing $a, b > 0$ and $\sigma(u) = 1/(1 + \exp(-u))$ is the sigmoid function. The final hydrogen bond loss is the cross-entropy between these predictions and the data,

$$\mathcal{L}_{\text{h-bond}} = \sum_{|i-j|>2} y_{ij}^{\text{data}} \log y_{ij}^{\text{model}} + \left( 1 - y_{ij}^{\text{data}} \right) \log \left( 1 - y_{ij}^{\text{model}} \right).$$

**Secondary Structure Prediction** We output standard 8-class predictions of secondary structure and score them with a cross-entropy loss.

### B.4 STABILIZING BACKPROPAGATION THROUGH TIME

The combination of energy function, simulator, and refinement network can build an atomic level model of protein structure from sequence, and our goal is to optimize (meta-learn) this entire procedure by gradient descent. Before going into specifics of the loss function, however, we will discuss a challenges and solutions for computing gradients of unrolled simulations in the face of chaos.

### B.5 CHAOS AND EXPLODING GRADIENTS

Gradient-based learning of iterative computational procedures such as Recurrent Neural Networks (RNNs) is well known to be subject to the problems of exploding and vanishing gradients (Pascanu et al., 2013). Informally, these occur when the sensitivities of model outputs to inputs become either extremely large or extremely small and the gradient is no longer an informative signal for optimization. We find that backpropagation through unrolled simulations such as those presented is no exception to this rule. Often we observed that a model would productively learn for tens of thousands of iterations, only to suddenly and catastrophically exhibit diverging gradients from which the optimizer could not recover - even when the observed simulation dynamics exhibited no obvious qualitative changes to behavior and the standard solutions of gradient clipping (Pascanu et al., 2013) were in effect. Similar phenomena have been observed previously in the context of meta-learning (Maclaurin et al., 2015) and are explored in detail in a concurrent work (Parmas et al., 2018).

In Figure 9, we furnish a minimal example that illustrates how chaos can lead to irrevocable loss of learning. We see that for even a simple particle-in-a-well, some choices of system parameters (such as too large a time step) can lead to chaotic dynamics which are synonymous with explosive gradients. This example is hardly contrived, and is in fact a simple model of the distance potentials between coordinates in our simulations. Moreover, it is important to note that chaos may not be easy to diagnose: for learning rates $\alpha \in [1.7, 1.8]$ the position of the particle $x$ remains more or less confined in the well while the sensitivities diverge to $10^{200}$. It seems unlikely that meta-learning would be able to recover after descending into chaos.

**The view per time step** Exploding gradients and chaotic dynamics involve the same mechanism: a multiplicative accumulation of sensitivities. In dynamical systems this is frequently phrased as 'exponentially diverging sensitivity to initial conditions'. Intuitively, this can be understood by examining how the Jacobian of an entire trajectory decomposes into a product of Jacobians as

$$\frac{\partial \boldsymbol{x}^{(T)}}{\partial \boldsymbol{x}^{(0)}} = \frac{\partial \boldsymbol{x}^{(T)}}{\partial \boldsymbol{x}^{(T-1)}} \frac{\partial \boldsymbol{x}^{(T-1)}}{\partial \boldsymbol{x}^{(T-2)}} \cdots \frac{\partial \boldsymbol{x}^{(1)}}{\partial \boldsymbol{x}^{(0)}}. \tag{10}$$

When the norms of the per time-step Jacobians $\frac{\partial \boldsymbol{x}^{(t)}}{\partial \boldsymbol{x}^{(t-1)}}$ are typically larger than 1, the sensitivity $||\frac{\partial \boldsymbol{x}^{(T)}}{\partial \boldsymbol{x}^{(0)}}||$ will grow exponentially with $T$. Ideally, we would keep these norms well-behaved which is the rationale recent work on stabilization of RNNs (Henaff et al., 2016; Chen et al., 2018b). Next we will offer a general-purpose regularizer to approximately enforce this goal for any differentiable computational iteration with continuous state.

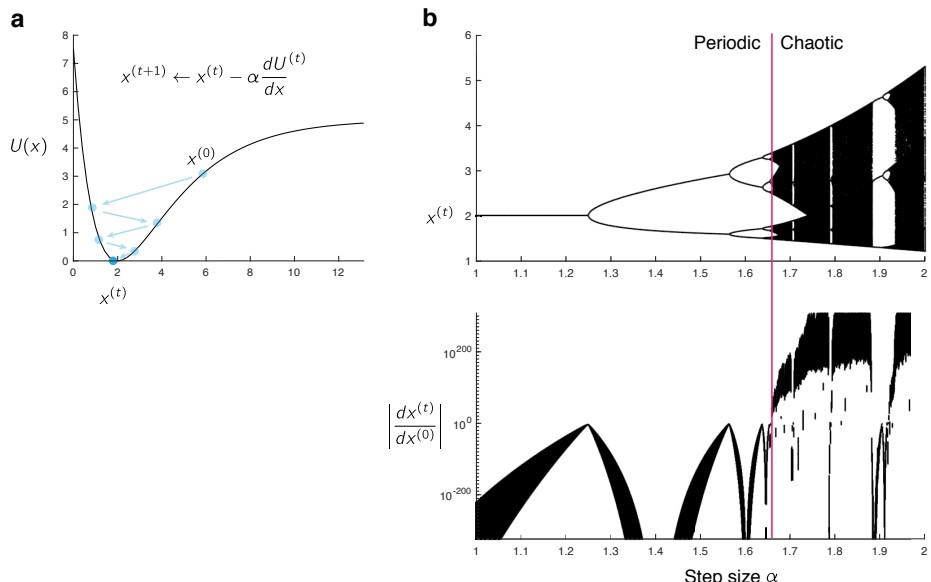

Figure 9: **Chaos impedes meta-learning for gradient descent in a well.** (a) Gradient descent of a particle in a well with initial conditions $x^{(0)}$ and step size $\alpha$. (b) Orbit diagrams visualize long-term dynamics from iterations 1000 to 2000 of the position $x$ (top) and the gradient $\frac{dx^{(t)}}{dx^{(0)}}$ (bottom). When the step size $\alpha$ is small, these dynamics converge to a periodic orbit over $2^k$ values where $0 \leq k < \infty$. After some critical step size, the dynamics undergo a period-doubling bifurcation (Strogatz, 2018), become chaotic, and the gradients regularly diverge to huge numbers.

**Approximate Lipschitz conditions**   One condition that guarantees that a deterministic map $F : \mathbb{R}^N \to \mathbb{R}^N$, $\boldsymbol{x}_t = F(\boldsymbol{x}_{t-1}, \theta)$ cannot exhibit exponential sensitivity to initial conditions is the condition of being non-expansive (also known as 1-Lipschitz or Metric). That is, for any two input points $\boldsymbol{x}_a, \boldsymbol{x}_b \in \mathbb{R}^N$, iterating the map cannot increase the distance between them as $|F(\boldsymbol{x}_a, \theta) - F(\boldsymbol{x}_b, \theta)| \leq |\boldsymbol{x}_a - \boldsymbol{x}_b|$. Replying the map to the bound immediately implies

$$|F^{(t)}(\boldsymbol{x}, \theta) - F^{(t)}(\boldsymbol{x} + \Delta\boldsymbol{x}, \theta)| \leq |\Delta\boldsymbol{x}| \tag{11}$$

for any number of iterations $t$. Thus, two initially close trajectories iterated through a non-expansive mapping must remain at least that close for arbitrary time.

We approximately enforce non-expansivity by performing an online sensitivity analysis within simulations. At randomly selected time-steps, the current time step $\boldsymbol{x}^{(t)}$ is rolled back to the preceding state and re-executed with small Gaussian perturbations to the state $\boldsymbol{\delta} \sim \mathcal{N}(0, 10^{-4}I)$[6]. We regularize the sensitivity by adding

$$\mathcal{L}_{Lyapunov} = \max\left(0, \log \frac{|F(\boldsymbol{x}^{(t)}) - F(\boldsymbol{x}^{(t)} + \boldsymbol{\delta})|}{|\boldsymbol{\delta}|}\right) \tag{12}$$

to the loss. Interestingly, the stochastic nature of this approximate regularizer is likely a good thing - a truly non-expansive map is quite limited in what it can model. However, being 'almost' non-expansive seems to be incredibly helpful for learning.

**Damped Backpropagation through Time**   The approximate Lipschitz conditions (or Lyapunov regularization) encourage but do not guarantee stable backpropagation. When chaotic phase-transitions or otherwise occur we need a fall-back plan to be able to continue learning. At the same time, we would like gradient descent to proceed in the usual manner when simulator dynamics

---

[6]For stochastic processes such as Langevin dynamics that depend on external noise, this must be cached and re-applied.

---

**Algorithm 4:** Damped Backpropagation Through Time

---

**Input** :Initial state $\boldsymbol{x}^{(0)}$, time-stepping function $F(\boldsymbol{x}, \boldsymbol{s}, \theta)$, external inputs $\boldsymbol{s}_1, \ldots, \boldsymbol{s}_T$
parameters $\theta$, Loss function $\mathcal{L}(\boldsymbol{x}_1, \ldots, \boldsymbol{x}_T)$, Damping factor $0 << \gamma < 1$
**Output** :Exponentially damped gradient $\nabla_\theta \mathcal{L}$
Initialize $\boldsymbol{x}^{(0)} \leftarrow \mathcal{F}(\boldsymbol{z}^{(0)})$;
**for** $t \leftarrow 2, \ldots, T$ **do**
    Compute time step $\tilde{\mathbf{x}}_t \leftarrow F(\boldsymbol{x}_{t-1}, \boldsymbol{s}_t, \theta)$;
    Decay the gradient$^\dagger$ $\boldsymbol{x}_t \leftarrow (1 - \gamma)\perp(\tilde{\mathbf{x}}_t) + \gamma\tilde{\mathbf{x}}_t$;
**end**
Compute loss $\mathcal{L}(\boldsymbol{x}_1, \ldots, \boldsymbol{x}_T)$ ;
Compute gradient $\nabla_\theta \mathcal{L} \leftarrow \texttt{AutoDiff}(\mathcal{L}, \theta)$ ;
$^\dagger$where $\perp(\cdot)$ is the `stop_gradient` function.

---

are stable. To this end we introduce a damping factor to backpropagation that can adaptively combat exponentially diverging gradients with exponential discounting (Algorithm 4).

Damped backpropagation can be seen as a continuous alternative to the standard approach of Truncated Backpropagation through Time. Rather than setting the gradient to 0 after some fixed intervals of time-steps, we decay it on the backwards pass of reverse-mode differentiation by a factor of $\gamma$. This is mildly evocative of the notion of discounted future rewards in reinforcement learning. During backpropagation this causes a biased estimate of Jacobians that favors short term sensitivities (or rewards) as

$$\partial \frac{\hat{\boldsymbol{x}^{(t)}}}{\partial \boldsymbol{x}^{(t-k)}} = \left( \left( \left( \gamma \frac{\partial \boldsymbol{x}^{(t)}}{\partial \boldsymbol{x}^{(t-1)}} \right) \gamma \frac{\partial \boldsymbol{x}^{(t-1)}}{\partial \boldsymbol{x}^{(t-2)}} \right) \cdots \gamma \frac{\partial \boldsymbol{x}^{(t-k+1)}}{\partial \boldsymbol{x}^{(t-k)}} \right) = \gamma^k \frac{\partial \boldsymbol{x}^{(t)}}{\partial \boldsymbol{x}^{(t-k)}}. \tag{13}$$

### B.6    Multiple Sequence Alignment Generation

We use multiple sequence alignments of evolutionarily related sequences for both profile construction (§ B.7) and (ii) data augmentation (§ B.8). For every domain in the dataset, we extracted the sequence from the PDB and then used jackhmmer (Eddy, 2011), to iteratively search the Uniprot90 database (Suzek et al., 2014) (release 4/2016) with 5 iterations and a length-normalized bitscore threshold of 0.3. We then removed sequences with over 50% gaps relative to the query sequence and then redundancy-reduced the alignment with hhfilter (Remmert et al., 2012) such that all sequences are at least a normalized Hamming distance of 0.8 away from one another.

### B.7    Profile Generation

We briefly describe how we construct evolutionary profiles, or position-specific scoring matrices (PSSMs), for each protein domain. Let $\mathcal{S} = \{\boldsymbol{S}^{(1)}, \ldots, \boldsymbol{S}^{(L)}\}$ be the set of $L$ columns of a multiple sequence alignment over $M$ sequences where each column $\boldsymbol{S}^{(i)}$ is an $M \times q$ matrix that one-hot encodes the sequence data at position $i$ (for an alphabet of size $q$). The regularized empirical frequency of letter $a$ at site $i$ is then

$$f_a^{(i)} = \frac{\alpha + \sum_j \boldsymbol{S}_{ja}^{(i)}}{\alpha + M},$$

where $\alpha$ is a pseudocount that we set to 10. We compute our PSSM features for letter $a$ at site $i$ as

$$w_a^{(i)} = \sigma \left( \log \frac{f_a^{(i)}}{B_a} \right)$$

where $\sigma(u) = \frac{1}{1+\exp(-u)}$ is the logistic sigmoid and $B_a$ is the average frequency of amino acid $a$ in UniProt (Apweiler et al., 2004).

Table 4: **Qualitative timings**. [†]Results on CATH dataset and 2 M40 GPUs.

| Method | Generation time | Training time |
|---|---|---|
| RNN baseline[†] | milliseconds | $\sim 1$ week |
| NEMO[†] | seconds | $\sim 2$ months |
| Coevolution-based methods | minutes to hours | Coupled to generation |
| Physical simulations | days to weeks | N/A |

### B.8 EVOLUTIONARY DATA AUGMENTATION

To reduce our reliance on alignments and the generation of profiles for inference of new sequences while still leveraging evolutionary sequence data, we augmented our training set by dynamically spiking in diverse, related sequence into the model during training. Given a set of $M$ sequences in the alignment we sample a sequence $t$ based on its normalized Hamming distance $d_t$ with probability

$$p_t = \frac{e^{\lambda_{\text{EDA}} d_t}}{\sum_{s=1}^{M} e^{\lambda_{\text{EDA}} d_s}},$$

where $\lambda_{\text{EDA}}$ is a scaling parameter that we set to 5. When the alternate sequence contains gaps, we construct a chimeric sequence that substitutes those sites with the query. This strategy increased the number of available sequence-structure pairs by several orders of magnitude, and we used it for both profile and 1-seq based training.

## C APPENDIX C: RESULTS

### C.1 STRUCTURE GENERATION AND PROCESSING

For each sequence from the CATH release 4.2 dataset, 100 structures were generated from both the profile and sequence-only models, while a single structure was generated from the RNN baseline models. The reported TM-scores were calculated using Maxcluster (Siew et al., 2000). A single representative structure was chosen from the ensemble of 100 structures using 3D-Jury (Ginalski et al., 2003). A pairwise distance matrix of TM-scores was calculated for all of the 100 structures in the ensemble. Clusters were determined by agglomerative hierarchical clustering with complete linkage using a TM-score threshold of 0.5 to determine cluster membership.

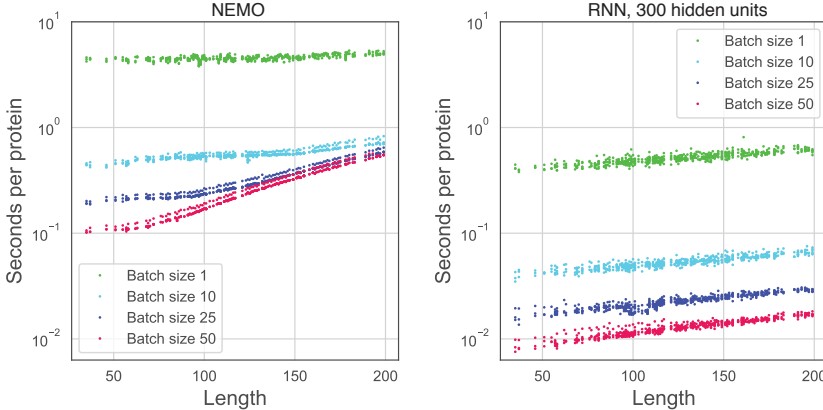

Figure 10: **Sampling speed.** Per-protein sampling times for various batch sizes across NEMO and one of the RNN baselines on a single Tesla M40 GPU with 12GB memory and 20 cores. For all results in the main paper, 100 models were sampled per protein followed by consensus clustering with 3D-jury, adding an additional factor of $10^2$ cost between NEMO and the RNN.

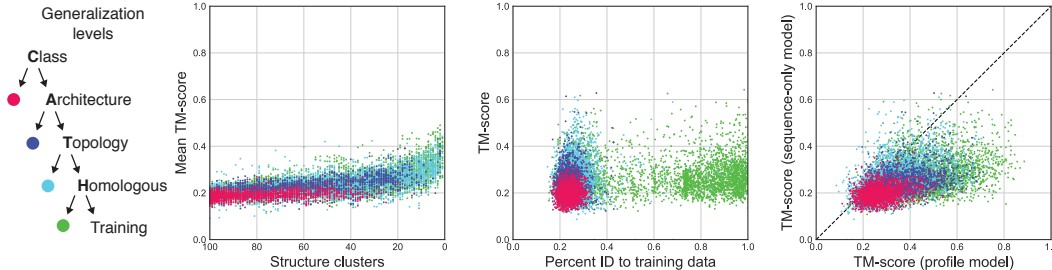

Figure 11: **Predictive performance of structures generated by the sequence-only model.** (left) Structures in the test set are hierarchically organized by CATH classification. Groups further up the tree are broader generalization. (center-left) Ensembles of models with increasing certainty tend to have a better average TM-score. (center-right) TM-score of 3D-jury-selected models versus distance from the training data. Withheld (right) Comparing the energy-based model with and without profiles. Profile information greatly improves protein model accuracy as judged by TM-score.

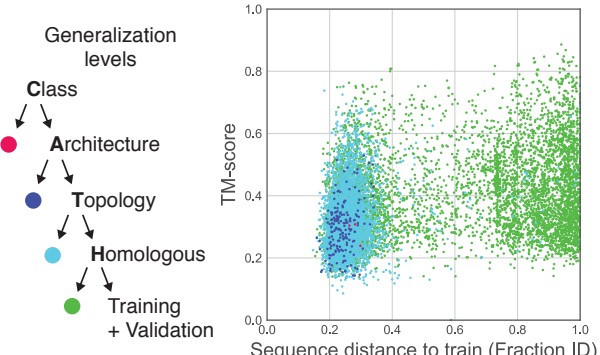

Figure 12: **Generalization results upon re-stratification.** Profile-based model.

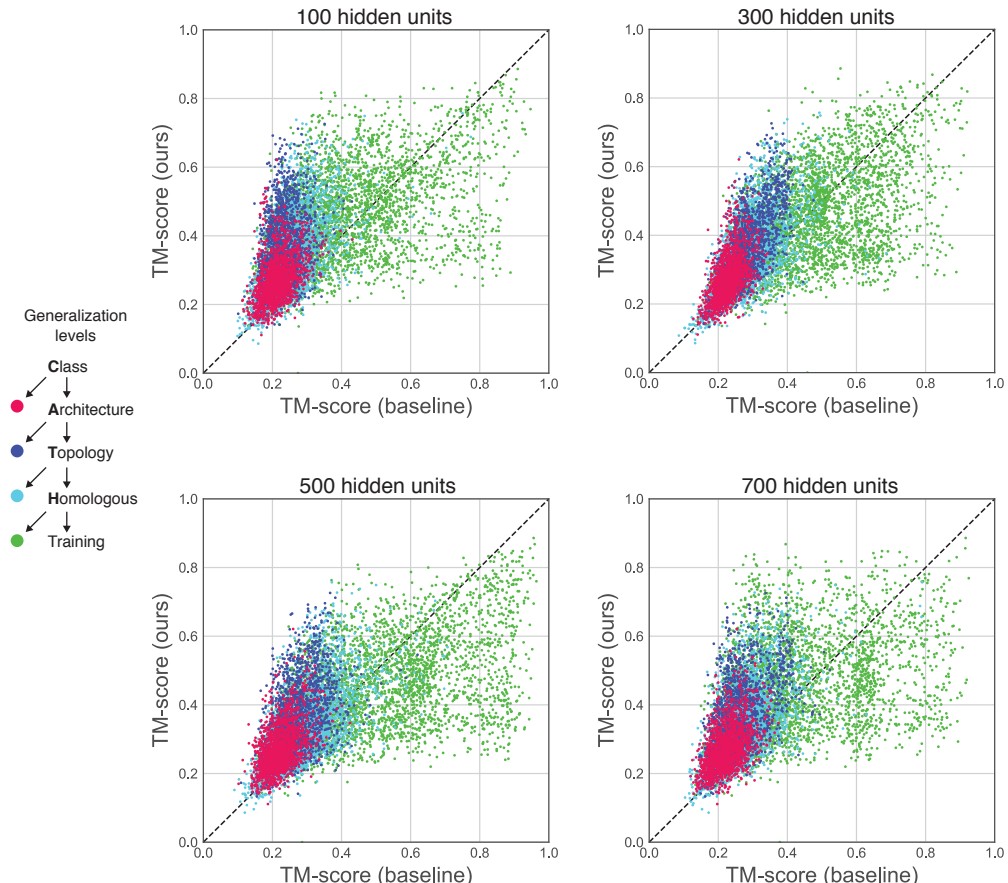

Figure 13: **RNN baseline performance for different hyperparameters.** Predictive performance of the two-layer bidirectional LSTM baseline models across a range of hidden unit dimensions compared to the energy model.

