# OpenReview forum: "Learning Protein Structure with a Differentiable Simulator"
_ICLR.cc/2019/Conference_

### Official Review · AnonReviewer2 · 2018-10-17

**Rating:** 6
**Confidence:** 3

**Review:**

Post-rebuttal revision: The authors have adressed my concerns sufficiently. The paper still has issues with presentation, and weak comparisons to earlier methods. However, the field is currently rapidly developing, and comparing to earlier works is often difficult. I believe the Langevin-based prediction is a significant and clever contribution. I'm raising my score to 6.

------

The paper proposes an end-to-end neural architecture for learning protein structures from sequences. The problem is highly important. The method proposes to use a Langevin simulator to fold the protein ‘in silico’ from some initial state, proposes numerous tricks for the optimisation, and proposes neural networks to extract information from both the sequence and the fold state (energy function). The system works on internal coordinates, which are conditioned and integrated on the fly. The method seems to perform very well, improving upon their baseline model considerably.

In spite of the paper being an outstanding work, I have two criticisms about the accessibility and impact of the paper on the broader ICLR audience. In its current form and complexity, the paper feels accessible mostly to a narrow audience.

First, the framework proposed in the paper is massive, containing a large amount of components, neural networks, simulators, integrators, optimisation tricks, alignments, profiles, stabilizations, etc. The amount of work done in the manuscript is staggering, but the method is also difficult to understand from reading the main manuscript alone. The 10+ page appendix is critical for understanding (for instance, the appendix reveals that MSA is used to generate more data), and even with it the method is difficult to grasp as a whole. This paper should be presented in a journal form with a presentation not hindered by page limits, while currently one needs to jump between the main text and appendix to get the whole picture. I also wonder if some parts of the system have already been published, and perhaps the presentation could be condensed that way.

Second, the introduction lists numerous competing methods both on the protein modelling side and on the MCMC vs optimisation side. The paper does not compare to any of these, which is strange, and makes it difficult to assess how much this paper improves upon state-of-the-art. Right now its unclear what is state-of-the-art in general. No bigger context of protein folding is given either, for instance, how well the method fares against purely alignment based approaches, or against purely physics-based simulators. Finally, the experimental section poorly describes how all the pieces of the system affect the final predictions. The discussion on the exploding gradients and dampening is excellent however. The only baseline is one with the simulator replaced by an RNN. There does not seem to be any running time analyses. As such, it is hard to interpret the current system, and it feels like a black box.

---

> ### Author Response · Authors · 2018-11-27
> **Response to AnonReviewer2**
>
> Thank you for your constructive review of this work. We hope that with improved presentation and contextualization, it can be relevant to a broad audience at ICLR.
>
> > In its current form and complexity, the paper feels accessible mostly to a narrow audience.
> > The amount of work done in the manuscript is staggering, but the method is also difficult to understand from reading the main manuscript alone. The 10+ page appendix is critical for understanding … This paper should be presented in a journal form with a presentation not hindered by page limits… I also wonder if some parts of the system have already been published, and perhaps the presentation could be condensed that way.
>
> While a journal might better accommodate the appendix, we believe that the complete system can be of interest to a general audience at ICLR because it connects recent interesting ideas in machine learning (e.g. differentiable simulators & meta-learning) to a challenging and well-known application domain with novel methodology (transform integrator, stabilization strategies, etc). To make these contributions more accessible, we have simplified the presentation of the model in the main text and added previously missing legends and overview paragraphs. (Lastly, in case its relevant, no parts of the model have previously been published.)
>
> > The paper does not compare to any of these, which is strange, and makes it difficult to assess how much this paper improves upon state-of-the-art. Right now its unclear what is state-of-the-art in general.
> … The only baseline is one with the simulator replaced by an RNN
>
> We focus on comparing to end-to-end approaches with a controlled dataset, largely due to the computational challenges of training the differentiable simulator model. We have added a discussion of Advantages and Disadvantages that more explicitly makes the connection to recent end-to-end methods for predicting protein structure in terms of angles (AlQuraishi et al)  and our baseline RNN. We focus on comparing to the RNN baseline because it shares the same loss and data augmentation strategies as our simulator, thus making clearer where differences in performance come from. While we do see that our differentiable simulator model can generalize more effectively to distant folds on our controlled dataset of ~35k proteins < 200AAs, the dataset splitting and significant cost of training (2 months on 2 GPUs) mean that it is difficult to evaluate the approach performance on large proteins. Nevertheless, new models in machine learning with better inductive biases but greater computational demands often get their start via medium-sized controlled datasets.
>
> > There does not seem to be any running time analyses
>
> Thanks for pointing this out. We have added a table of qualitative running times in the results section.
>
> > the experimental section poorly describes how all the pieces of the system affect the final predictions
> > it feels like a black box.
>
> While we could not afford to perform ablation studies of individual components given the long training time, we believe that structured nature of a differentiable simulator can make it easier to interpret and engineer than purely neural architectures. For example, the Markov Random Field formulation of the energy function means the sequence and structure features can be interpreted separately, and both the efficacy of Langevin dynamics and benefits of alternative coordinate parameterizations to sampling are well-understood phenomena.

---

> > ### Comment · AnonReviewer2 · 2018-11-27
> > **Still concerns about comparisons**
> >
> > Thanks for your response.
> >
> > I am still concerned by lack of comparisons to earlier methods. As described very nicely in the introduction, the main energy-based method to compare are Jumper and Krupa, while AlQurashi and Anand are the competing deep models. Both AlQurashi and Anand are very recent, and there does not seem to be a public manuscript of the Anand yet, hence comparing to them would be unreasonable. However, both Jumper and Krupa models seem simple enough to be implemented and compared to (neither seem to have implementations available). The Jumper's method claims 3 days of computations per protein, and in that case the key characterisation would be accuracy vs speed of different methods.
> >
> > Furthermore, In AlQurashi they explicitly compare to CASP 7-12 top methods, which seem to be the classic conventional methods Rosetta, I-TASSER, HHPRED, etc. They also seem to recreate the conditions of earlier CASPs to get comparable results, which is a great idea.
> >
> > One needs to lay out the baselines and compare to earlier methods, especially when one is proposing a novel paradigm (deep learning). Using only RNN as baseline is not sufficient to evaluate how the proposed method performs in different cases. You claim to focus on "comparing to end-to-end approaches", however there seem to be none currently except for the somewhat artificial RNN baseline. CASP methods or standard protein servers as baselines are then necessary to show that the new (deep) paradigm has merit. I would also encourage to authors to check the developments in CASP13. I think comparisons are the main issue with the paper, and needs to be properly addressed.
> >
> > I'm also not sure how the RNN baseline shows "where differences in performance come from". Can you explicate what parts of NEMO cause its performance to be better than RNN, and why?
> >
> > The running times could be more specific by detailing processing time required per protein.
> >
> > On the presentation, I think the paper mostly needs a "big picture" figure that shows how the neural networks play into the simulation and energy systems. For instance, AlQurashi's figs 1+2 are very illuminating in this respect.

---

> > > ### Author Response · Authors · 2018-11-28
> > > **Clarity on framing of work**
> > >
> > > We apologize and feel that we may have been unclear in how we are framing this work. Our primary goal is to explore the utility of a learned, differentiable simulator in the context of the challenging problem of protein folding and to show that it can have better inductive bias than an approach based directly on angle-prediction (i.e. AlQuraishi and our baseline RNN). To be clear, we do not expect that this method would be competitive at CASP (where all forms of sequence and structure data before a certain date are fair game), as the cost of scaling training limited us to a medium-sized dataset of proteins of limited length (200 AAs) from which many topologies and architectures have additionally been held out.
> > >
> > > We feel that it is still useful to present this novel methodology before it can scale to the challenge of CASP, because we are trying to argue about the inductive bias of simulators and not to make a general claim about deep methods versus conventional methods. Both our method and AlQuraishi’s can create models with hundreds to thousands of atoms in milliseconds to seconds, which is far off from the timescales of physics based approaches (even the very impressive work of Jumper et al measures simulation times on the order of CPU-days), and we consider that sufficiently interesting to motivate research on ‘deep’ approaches.
> > >
> > > With that in mind, we believe that the RNN baseline is a meaningful comparison and not ‘artificial’. Within the very recently emerging field of end-to-end models of protein structure, the idea of directly predicting internal coordinates (AlQuraishi) may be considered the other established paradigm (In Anand et al’s currently available manuscript, they focus on structural imputation & generation rather than prediction). Like AlQuraishi’s work, our RNN baseline composes a multilayer, bidirectional LSTM that predicts internal coordinates (in our case, coarse) with a scoring function on the resulting atomic Cartesian structure (in our case, after imputation). If we were to directly retrain the AlQuraishi model on our dataset, there are many possible explanations for performance differences such as different losses, the imputation network, training details, etc. We designed the experiments around our baseline because it allowed us keep those factors constant while replacing only the simulator portion of the model (this is what we meant by “where differences in performance come from”). Regarding the methods of Jumper and Krupa, we certainly want to acknowledge their contributions and related ideas but find that the costs of training (much longer simulations per protein) would be very challenging to scale to our dataset of 35k proteins.
> > >
> > > To your other suggestions, we will add per-protein running time statistics and specific schematics for the 1D and 2D neural network modules upon revision.
> > >
> > > We thank you for your patience with this work, and hope that the ICLR community can find value in the methodological and conceptual contributions.

---

### Official Review · AnonReviewer1 · 2018-10-29
**Interesting idea with some weaknesses in the evaluation**

**Rating:** 7
**Confidence:** 3

**Review:**

This paper presents an end-to-end differentiable model (NEMO) for protein structure prediction. I found this paper very interesting and the idea of training the network through the sampling procedure promising. The authors present the challenges and techniques (damping, Lyapunov regularization etc) in detail.

The paper is clearly written, however the description of the method can be confusing. This stems in part from the many components of the network as well as the fact that the protein is represented using various coordinate systems and features, so that it is not easy to follow which applies at each stage. Fig. 6 in the appendix helps, however it would be better to have a (perhaps more concise) overview in the main text.

In the evaluation, the NEMO method is compared to a baseline approach using RNNs. While NEMO trained on profile features performs best, the baseline is trained on sequences only. However, it outperforms the NEMO model trained on sequence-only in every category. Therefore, it would be interesting to see whether NEMO outperforms a baseline trained on profile features. Otherwise, I am not certain whether I can follow the conclusion that "NEMO generalizes more effectively". Beyond that, it would be interesting to see some generated atomic substructures from the imputation network, in particular an analysis of how diverse the generated atom positions are and whether they depend on the local environment.

Overall, I appreciate the general idea and find the proposed approach very interesting. The contribution could have been stronger with a more detailed evaluation and better presentation.

---

> ### Author Response · Authors · 2018-11-09
> **Quick clarification: RNN baselines are also trained on profiles**
>
> Thank you for your review and positive words about the idea and approach. While we will respond in full later, we wanted to briefly clarify that all RNN models and NEMO results of Fig 4,5 were trained on profiles. The sentence "We report the results of a sequence-only model in Table 1 and Figure 4" is a figure-link typo and should instead read "We also report the results of a sequence-only NEMO model in Table 1 and Figure 9." We apologize for the confusion and will make these points clearer upon revision.
>
> In the meantime, we hope that this can clarify that our main claim about generalization is based on comparing profile-based NEMO to profile-based baselines.

---

> ### Author Response · Authors · 2018-11-27
> **Response to AnonReviewer1**
>
> Thank you for your positive comments and constructive suggestions.
>
> > The paper is clearly written, however the description of the method can be confusing. …
> Fig. 6 in the appendix helps, however it would be better to have a (perhaps more concise) overview in the main text.
>
> To improve accessibility, we have simplified explanations of individual components and added overviews to the main text and Figure 6.
>
> > it would be interesting to see whether NEMO outperforms a baseline trained on profile features
>
> While it was not clear from the text, the original RNN was indeed trained on profiles, and we apologize for the confusion (Please see our previous comment).
>
> > it would be interesting to see some generated atomic substructures from the imputation network, in particular an analysis of how diverse the generated atom positions are and whether they depend on the local environment.
>
> We agree that this is an interesting question, though our current model will likely not give an interesting answer since the imputation network is a deterministic mapping (ignoring dropout). That said, all secondary structure calls in the visualizations come directly from hydrogen bonding calls in pymol (default settings) which suggests that model can capture some aspects of (locally) orientation-dependent atom placement.

---

### Official Review · AnonReviewer4 · 2018-11-11

**Rating:** 7
**Confidence:** 5

**Review:**

Overall this is an important piece of work that deserves publication at ICLR. I recommend to the authors revise their manuscript to make it more accessible to the machine learning community and that they provide better context to allow them to assess the relative quality of the work compared to state of the art results.

# Quality

The hypothesis that the authors set out to resolve is whether there is an advantage in using an energy function sampled by Langevin dynamics versus simply using a neural network to regress shape from sequence. They construct a flexible deep energy model where the sequence and structure dependent parts are separated in such a way that fast rollouts are possible. They also adapt the learning algorithm to  ensure that long rollouts can be carried out and present a clever trick for integrating internal coordinates efficiently on a GPU.

The only criticism in terms of quality of work is that it somewhat lacks putting in context with results from the larger community, for example how well does the model compare in terms of speed and accuracy with co-evolutionary approaches? I realise it will not be possible to give a completely fair like to like comparison, but it will help readers put the results in context if they understood, for example, what the average TM score for CASP12 results was, as summarized in this paper for example: https://onlinelibrary.wiley.com/doi/full/10.1002/prot.25423. Similarly, it would be useful to compare the baseline - at least qualitatively - with the results from AlQuraishi et. al. whose model seems very similar in spirit.

# Clarity

I think in terms of clarity, the paper could be improved a little to take into account the audience of ICLR. In particular:

* It may be useful to add a sentence of how profiles have been found to improve secondary structure prediction greatly. Currently the text makes it sound as though they constitute a sort of 'data augmentation', whereas in my opinion they add information compared to the sequence alone. In fact a brief explanation of the importance of homology might help the reader understand the relevance of the hierarchical approach taken to splitting the training set.

* Fig. 2 caption. Could add some information to explain what panel B is showing. I think this would go a long way to explain why both cartesian and internal coordinates are important.

* Fig. 4 second panel. The x axis should be labeled fraction or be numbered 0-100.

* Fig 4. caption. The figure does not have a caption explaining what the graphs are showing. This would be a good place to explain that the colors refer to test sets that overlap with the training set in the full CATH code (black), overlap only in the CAT code (orange) etc. I admit I had found the explanation of the test/train/validation split rather confusing. It is not clear what the validation set is used for, i.e. which hyper-parameters have been tuned on it etc.

* The nature of the loss. The appendix does a good job in describing each term in the loss function, but does not explain how the empirical loss function and the log-likelihood terms are mixed together.

# Originality

The work is original and is references the relevant literature.

---

> ### Author Response · Authors · 2018-11-27
> **Response to AnonReviewer4**
>
> We thank the reviewer for the positive comments about the approach and suggestions for how to improve the presentation.
>
> > It may be useful to add a sentence of how profiles have been found to improve secondary structure prediction greatly. Currently the text makes it sound as though they constitute a sort of 'data augmentation', whereas in my opinion they add information compared to the sequence alone.
>
> We agree that evolutionary profiles add far more information than data augmentation, and have added an explicit point of comparison in the results section to draw the connection to SS prediction.  We have also clarified the distinction between data augmentation and profiles in Appendix B.
>
> > lacks putting in context with results from the larger community, for example how well does the model compare in terms of speed and accuracy with co-evolutionary approaches? … Similarly, it would be useful to compare the baseline - at least qualitatively - with the results from AlQuraishi et. al. whose model seems very similar in spirit.
>
> We agree that the paper needs to provide better context in the landscape of methods for protein structure prediction and have tried to address this by adding an ‘Advantages and Disadvantages’ paragraph to the Results. Since scaling our method to larger datasets of proteins remains difficult with current computational resources, we focus primarily on comparing to other end-to-end approaches, of which an RNN-based angle prediction (of AlQuraishi et al) is the other major approach. Hopefully our updated text and figure legends can clarify this.
>
> > [Fig 2., Fig 4 second panel and caption]
>
> Thank you for these suggestions. We have fixed these figure legends to be clearer.
>
> > does not explain how the empirical loss function and the log-likelihood terms are mixed together.
>
> Thanks for pointing this out. Our loss involves a simple sum of all terms without weights, and we have added a sentence to clarify this.
>
> >I admit I had found the explanation of the test/train/validation split rather confusing.It is not clear what the validation set is used for, i.e. which hyper-parameters have been tuned on it etc.
>
> We have improved the explanation of how we split the dataset hierarchically and temporally to capture different generalization difficulties. We did not explicitly tune hyperparameters on the validation set (in part due to the long training time, for which 200k iterations was what we could afford), but we did allow ourselves to look at the validation set during model development and thus refer to it as such.

---

### Official Review · AnonReviewer3 · 2018-11-12
**Interesting paper - not clear and mature enough**

**Rating:** 6
**Confidence:** 5

**Review:**

The paper proposes a new end-to-end training framework for computational prediction of protein structure from sequence.
This is a very important problem and any progress due to new data and/or methods for utilizing may have high impact.

The paper presents several technical contributions in the modelling and training procedure - for example, automatic transformation between Cartesian and angular coordinates, using Langevin dynamics, and imputation method to get fine atomic coordinates.

The overall breadth and depth of the methods presented in the paper are impressive. The paper describes a quite complicated systems with multiple modules interacting between them. The paper doesn't describe the system built in enough details, although many of the details are given in the appendix.
Figure. 6 presents a scheme of the entire system, but it lacks details about the different modules, and it is not clear how they interact and how their training together is performed.
The pseudo-code boxes describing Algorithms 1-4, and Table 2 describing the representation are informative and helpful, and more descriptions of this type would help.
For example:    - In Algorithm 3, what do 'CartesianStep' and 'ClippedInternalStep' mean? where are they described? (should have their own boxes/description).
		- I didn't see an Algorithm describing the atomic imputation part.
		- It would be good to add a high-level pseudo-code for the entire end-to-end training algorithm. In it there could be calls to Algorithms 1-4 when needed.

There is also no single place where all the parameters used by the authors to achieve their empirical results are presented
(e.g. learning rates, Gaussian kernel widths, how are random time steps for enforcing Lipschitz condition chosen etc.).
In addition, the empirical loss defined in eq. (8) is a sum of 6 different losses. It is not clear how are these very different losses scaled to the same 'units', which ones are more important,
if and how are constants multiplying them chosen to give lower/higher weights to some of the losses etc. - I guess these choices will have a large effect on the training.

The authors present generalization results of their trained model in predicting 3D structures from CATH at different generalization level
(i.e. different similarity levels to the training set proteins). It is not clear to me how good are these results, except that they are shown
to be better than a baseline simple model. How well does the author's model compare to other recently suggested end-to-end models?
(the authors mention AlQuraishi, Anand&Huang, papers). How do they compare to state-of-the art structure prediction programs? (e.g. CASP winners)?
I realize giving an automatic end-to-end solution is interesting even if performance is below that of best programs, but still it would be good to know gaps.
If such comparisons are less meaningful/not practical to perform this should be argued convincingly.
It would also be useful to add some metrics of running time - it is not clear how computationally heavy and scalable is the author's model and training, compared to other methods.


There are many typos and inconsistent notations which makes it harder for the reader to understand the paper.
For example, 'Figure ??' in multiple locations, wrong Figure referenced, using s vs. S for sequence - S is defined as an L*20 matrix but in the appendix there are
3 indices: s_{i,l,j} and it looks like different sequences in alignment should be denoted s_i.
Equation for M_{l,j} isn't clear: j is used both as fixed index and index in summation.
The indexing in 'orientation vectors' v-hat_ij definition seems off (the formula of base vectors gives 0/0)

---

> ### Author Response · Authors · 2018-11-27
> **Response to AnonReviewer3**
>
> We thank the reviewer for the extensive comments as well as suggestions for improving the presentation and evaluation.
>
> >Figure. 6 presents a scheme of the entire system, but it lacks details about the different modules, and it is not clear how they interact and how their training together is performed.
>
> We apologize for the lack of clarity and have added a legend to this figure that walks through the complete sampling process (which is, in turn, backpropagated through).
>
> > The pseudo-code boxes describing Algorithms 1-4, and Table 2 describing the representation are informative and helpful, and more descriptions of this type would help.
> >  what do 'CartesianStep' and 'ClippedInternalStep' mean? where are they described?
>
> We are glad that these algorithm boxes are helpful, and while we have not made them for 'CartesianStep' and 'ClippedInternalStep', these computations refer to the Langevin Dynamics and Speed Clipping paragraphs in Appendix A.
>
> > I didn't see an Algorithm describing the atomic imputation part.
>
> The atomic imputation was small and potentially easy to miss, but is defined in Section 2.4 ‘atomic imputation’. We have modified the formatting and added an overview paragraph to highlight its importance.
>
> > There is also no single place where all the parameters used by the authors
>
> While the complexity of the model makes presenting the hyperparameters in table form cumbersome, we intend to release the code which includes hyperparameters as structured objects.
>
> > if and how are constants multiplying them chosen to give lower/higher weights to some of the losses etc
>
> Regarding the loss, we simply sum the individual loss terms and have not explored weighting (owing to the the costly training time). We have clarified this in the text.
>
> > It is not clear to me how good are these results, except that they are shown to be better than a baseline simple model. How well does the author's model compare to other recently suggested end-to-end models? (the authors mention AlQuraishi, Anand&Huang, papers). How do they compare to state-of-the art structure prediction programs? (e.g. CASP winners)?
> I realize giving an automatic end-to-end solution is interesting even if performance is below that of best programs, but still it would be good to know gaps.
> If such comparisons are less meaningful/not practical to perform this should be argued convincingly.
>
> To better contextualize our model, we have added an Advantages and Disadvantages discussion as well as an improved explanation of the baseline. The RNN baseline method is similar to the approach of AlQuraishi (though differing in the use of coarse-to-fine reconstruction as well as our loss terms). We focus on comparing to the baseline model because it uses the same loss and imputation network, thus isolating the differences to the simulator itself. Regarding CASP: Although our method was able to scale to training on a database of ~35k protein domains up to length 200 (on 2 GPUs & 2 months), this particular dataset excludes the longer proteins and more diverse templates that would be necessary to be relevant to CASP.
>
> > It would also be useful to add some metrics of running time
>
> We have added a qualitative table of approximate running times for our methods as well as conventional protein folding approaches.
>
> > typos and inconsistent notations
>
> Thank you for pointing these typos out, we believe they are now fixed.

---

### Public Comment · (anonymous) · 2018-10-02
**Confused about testing and interpretation of results**

I've really enjoyed reading your paper, but I'm left very confused about the testing procedures and the quality of the results obtained. The key issue when reviewing papers on protein structure prediction is whether or not there is any overlap between the training/validation data sets and testing data. Here it seems that you have tried to be very stringent about this by splitting your dataset of domains along CATH boundaries - so two domains with different CATH codes should in theory be unrelated and thus have no detectable sequence similarity. Likewise two proteins with different CAT codes should have different folds, CA different architectures and C having different classes. Having trained and validated on the domains in CATH release 4.1 you then tested on domains in CATH release 4.2, but only those that are unrelated to the training set (different CATH numbers) or have different folds (different CAT numbers). Assuming that was done without bugs, then I find it hard to understand the middle panel of Figure 4. Unless I’m misunderstanding what’s going on there (the x-axis scale is wrong by the way), it seems to suggest that a large proportion of your testing set is actually quite similar to your training set. In some cases identical (100% identical protein sequence). Does that not indicate that there is sequence overlap between training and test data. Of course it’s impossible to know whether the overlap is with the validation set rather than the training set, but that would still be problematic.

If those highly similar sequences were included in the statistics shown in Table 1, for example, it would make the results there very difficult to interpret.

Looking at some of the specific examples of folded domains shown, it would have been useful to know what the sequence similarity is between the target and the most similar protein domain in the training/validation set. For example, I note that domain 2oy8A03 shares 100% sequence identity with domain 3ckgA02 (one is simply a deletion mutant relative to the other), which was already present in CATH release 4.1 and so must have been either in the training set or at least the validation set. If this is true, then the network has simply recapitulated what it has already seen and hasn’t actually predicted anything. Other examples shown have similar issues e.g. 4ykaC00, which is identical to 2wa9B00 in CATH 4.1 and indeed has the exact same CATH code, and so I don’t think should be included in the test set at all. Probably I have just misunderstood the exact way you’ve effected your training/test split, so I’d welcome any clarification you can give.

---

> ### Author Response · Authors · 2018-10-02
> **To clarify: Training set is hierarchically purged**
>
> Thank you for your question about the training and test splits; we will do our best to clarify briefly here as well as update the manuscript at the next opportunity.
>
> We very much agree that careful analysis of test and train overlap is one of the key issues when interpreting the results on protein structure prediction and we created our dataset in an attempt to frame this problem in terms of widely used (ie CATH) fold classifications.
>
> To clarify we *did not* train on all of CATH 4.1 but rather intentionally hierarchically purged the training set at multiple levels of A, T and H. So, after collecting all folds in 4.1 (subject to max length 200 and class 1,2,3), we then randomly selected a subset of A classifications and purged all folds descending from these into the A-level validation set. We repeated this process two more times for T and H. While the specific domain-level splits are contained in a large table not feasible for attaching to an openreview submission, we intend to make these available and will try to summarize the high level held out classifications at next update to the manuscript.
>
> Regarding the middle panel of Figure 4, the x axis is correct and our test set from CATH 4.2 does (naturally) contain some folds of very high (sometimes complete) sequence overlap. However, because we purged the training set (as just described above), the test set also contains many sequences of low/random sequence overlap (left cluster in middle panel) and low classification overlap. The color coding on this scatter plot indicates how close the given test domain is to the training set, where blue means that the ATH classifications were not present in train (column C in table 1), magenta means TH were not present in train (column A), orange means H was not present in train (column T) and gray (column H) means that the full CATH classification *was* present in train. The showreel plot contains only folds from the A and T columns (magenta and orange).
>
> We hope that hierarchical purging approaches to evaluation such as this will be more widely used in the future because they allow testing fold generalization more systematically across thousands of domains (rather than only doing temporal purging).
>
> To conclude, our main claim is that the model is able to (sometimes) produce reasonable (TM>0.5) predictions for these difficult ATH, TH, and H problems created by our purging process and that it does so more accurately than a baseline that predicts angles directly without a folding process.

---

> > ### Public Comment · (anonymous) · 2018-10-02
> > **Still a bit unclear on the purging**
> >
> > Thanks for the clarifications.
> >
> > Minor point, but that Fig. 4B x-axis is definitely incorrect. The scale goes from 0-1 and the axis label states that the units are % sequence ID (i.e. the maximum value is 100).
> >
> > "Hierarchical purging" like that is commonly practised in the comp. biol. community but it's unclear how that purging process has been extended to the _test_ set, which is based on CATH 4.2. I understand that you have split your _train_ and _validation_ sets according to a randomly selected subset of C, CA, CAT numbers found in CATH 4.1 - that's fine - I get that bit - but after training with that validation set, you cannot then have the same purged CAT numbers present in your test set because you will have fitted your model to your training set and then selected a model which does well on the domains in the validation set.
> >
> > In reality you need to produce three splits of the CATH classification - train, validation and test. The test CAT codes should not have been used for either training or validation.  Possibly this is exactly what you did, but it really could do with some clarification in the text.
> >
> > However, if that is what you did, then I don't understand how you can end up with the plot in Fig. 4B - because that would suggest that you have a lot of protein domains which have _different_ topologies (T in CATH) but which still have very high sequence similarity. That just can't be right. There are virtually no known examples of proteins with high sequence similarity which do not have the same topology. Even a sequence identity of just 30% is typically enough to guarantee that the two structures will have the same topology (TM-score > 0.5).
> >
> > From what you say, though,  it sounds like your test set was simply all the new domains added to CATH 4.2, and that some happen to overlap with the training set domains and some don't. That would explain what I see in Figure 4B alright, but it would pretty much invalidate your final results as your test set would be contaminated. Surely you don't mean that?
> >
> > Sorry to bang on, but if it wasn't such a potentially interesting paper I wouldn't care enough to ask!

---

> > > ### Author Response · Authors · 2018-10-17
> > > **Test set is stratified**
> > >
> > > Thanks for catching the fraction (0-1) / percent (0-100) mislabeling, we will fix it.
> > >
> > > It is correct that we split train and validation based on CATH 4.1 (hierarchical split) and tested on everything that was new to CATH 4.2 (temporal split), with the test set stratified into subsets of varying difficulty, from the very easy to the very difficult, and clearly labelled as such. While we currently stratify by CATH similarity between {test} and {train} (to evaluate more fold types), we can also include results that stratify by CATH similarity between {test} and {train+validation} (at the cost of reduced fold diversity). Since we subject all models and baselines to the same splits either way, all of these are interpretable measures of generalization.

---

### Meta-Review · Area_Chair1 · 2018-12-12
**Novel end-to-end-differentiable model for protein structure prediction**

**Confidence:** 4
**Recommendation:** Accept (Oral)

**Metareview:**

This paper presents a differentiable simulator for protein structure prediction that can be trained end-to-end. It makes several contributions to this research area. Particularly training a differentiable sampling simulator could be of interest to a wider community.

The main criticism comes from the clarity for the machine learning community and empirical comparison with the state-of-the-art methods. The authors' feedback addressed a few  confusions in the description, and I recommend the authors to further polish the text for better readability. R4 argues that a good comparison with the state-of-the-art method in this field would be difficult and the comparison with an RNN baseline is rigorously carried out.

After discussion, all reviewers agree that this paper deserves a publication at ICLR.